# Disentangling edge and bulk spin-to-charge interconversion in MoS$_2$ monolayer flakes

Rodrigo Torrão Victor [1], Syed Hamza Safeer [1,2], John F. R. Marroquin [3], Marcio Costa[4], Jorlandio F. Felix [3] ✉, Victor Carozo [5], Luiz C. Sampaio [1] & Flavio Garcia [1] ✉

Semiconductor transition metal dichalcogenides are an archetype for spintronic devices due to their spin-to-charge interconversion mechanisms. However, the exact microscopic origin of this interconversion is not yet determined. In our study, we investigated light-induced spin pumping in YIG/MoS$_2$ heterostructures. Our findings revealed that the MoS$_2$ monolayer microsized flakes contribute to spin current injection through two distinct mechanisms: metallic edge states and semiconductor area states. The competition between these mechanisms, influenced by the flake size, leads to different behaviors of spin-pumping. Our calculations of the local density of states, by means of density functional theory, of a flake show that light-driven spin current injection can be controlled based on the intensity of light with a suitable wavelength. We demonstrate that a lightdriven spin current injection can enhance up to very high values, attenuate, or even switch on/off the spin-to-charge interconversion. These results hold promise for developing low energy-consuming opto-spintronic device applications.

Spintronics has emerged as a promising field for developing next-generation devices that intertwine the spin and charge degrees of freedom[1–3]. Research shows that the spin Hall effect (SHE) in metals with high spin-orbit coupling is a key effect on the interconversion of spin and charge currents[4–7]. However, other effects might be even greater than the SHE. Systems with orbital degrees of freedom can exhibit a significant orbital Hall effect (OHE)[8]. Materials with a non-zero Berry curvature are capable of generating spin accumulation, termed as valley Hall effect (VHE)[9–11]. In addition, the Rashba-Edelstein effect (REE), stemming from the breaking spatial inversion symmetry at interfaces, can lead to orbital and spin currents[12–15]. Generally, these effects are called Hall effects.

One of the main challenges in practical applications of spintronics is the energy efficiency control of magnetic and spin states and the interconversion between charge and spin currents, especially at ultra-fast time and small scales. In this scenario, light can play a pivotal role as an energy-efficient way to overcome these issues. However, several challenges still need to be addressed, such as optimizing the efficiency of light-induced magnetic effects, improving the stability of materials, and designing scalable spintronic light drive devices[16]. A deeper understanding of the fundamental interaction of light with spintronic materials is crucial for actual applications of opto-spintronic devices.

The 2-dimensional (2D) transition metal dichalcogenides (TMDs), especially semiconductors with hexagonal structures, stand out as an excellent investigation archetype to explore the fundamental origins of spin-to-charge interconversion and the influence of light. Encompassing a distinctive set of features, TMDs can manifest various phenomena, including SHE, OHE, VHE, and REE. Among the TMDs, molybdenum disulfide (MoS$_2$) is particularly interesting. A MoS$_2$ monolayer comprises two S layers sandwiching one Mo layer in a covalently bonded hexagonal structure. This structure leads to a semiconductor state in the bulk and broken inversion symmetry,

[1]Centro Brasileiro de Pesquisas Físicas, R. Dr. Xavier Sigaud, 150, Urca, Rio de Janeiro 22290-180 RJ, Brazil. [2]Materials Science Laboratory, Department of Physics, Quaid-i-Azam University, Islamabad 45320, Pakistan. [3]Institute of Physics, LabINS, University of Brasília (UnB), Brasília, DF 70910-900, Brazil. [4]Instituto de Física, Universidade Federal Fluminense, Niterói 24210-346 RJ, Brazil. [5]Departamento de Física, Pontifícia Universidade Católica do Rio de Janeiro, Rio de Janeiro 22451900 RJ, Brazil. ✉e-mail: jorlandio@unb.br; fgarcia@cbpf.br

resulting in a combined spin-momentum locking and high spin-orbit coupling (SOC) required for the Hall effects[17–19]. MoS$_2$ monolayers can exhibit different edge states depending on their termination. When grown in a sulfur-rich environment, the flakes of MoS$_2$ monolayer take on triangular shape and zigzag edge terminations[20–25]. This specific termination results in the emergence of metallic edge states, which have been both experimentally observed and theoretically calculated in a narrow region a few nanometres wide along the edges of triangular flakes[26–30]. Hence, the semiconductor states in 2D are expected to coexist with the metallic states located at the edges of the flakes. As a result, the edge and the 2D semiconductor area states can contribute differently to the spin-to-charge interconversion through Hall effects and its Kelvin-Onsager thermodynamic reciprocal effects.

Several experiments have been recently conducted to measure the spin-to-charge current conversion in MoS$_2$. These experiments were carried out on various MoS$_2$ systems, such as a single flake of monolayer[31,32], large areas[33], several monolayered flakes[34], a few layers thick flakes[35,36], and the dependence of its thickness[37]. These studies provide insights into the fundamental physics of spin transport in MoS$_2$ based devices. However, since all contributions were observed simultaneously in those works, it was impossible to disentangle the different edge and area contributions.

One of the most common approaches to analyze the spin-to-charge interconversion is to measure the broadening of the ferromagnetic resonance (FMR) linewidth spectrum of soft ferromagnetic materials in contact with the material under investigation. This process is known as spin pumping, and it occurs due to the injection of angular momentum from the magnetization precession of the ferromagnetic layer to the MoS$_2$ layer, allowing controlled probing of a material's spin current injection. Since the spin current can only be injected in the TMDs in direct contact with the ferromagnetic layer, this technique can be used to probe small areas, such as tiny flakes, as well as large areas, such as big or abundant flakes[38]. Spin pumping is a method that has the advantage of averaging the flakes and reducing the impact of local defects in comparison to localized techniques like Kerr microscopy, spin transfer torque, and electrical measurements, which examine only one flake at a time[39–47]. Moreover, spin pumping does not need electrical contacts in the sample, which could suppress some contributions from Hall effects. This is particularly important because of the differences in conductivity between the 2D semiconductor and the metallic edge states present in the MoS$_2$ flakes. In addition, if the material used to inject the angular momentum is a magnetic insulator, such as yttrium iron garnet (YIG), it injects a pure spin current, avoiding unwanted electrical effects[48,49].

Although significant efforts have been made to develop new tools and theories to better understand TMD spintronics, a study that disentangles the fundamental contributions of the Hall effects and the influence of light has yet to be explored. In this context, we conducted a spin pumping study on the enhancement of YIG's Gilbert damping, which revealed a competitive interplay between two distinct spin pumping channels: one arising from the metallic edge states and the other originating from the 2D semiconductor area states. This competition was modulated by varying the aspect ratio of MoS$_2$ monolayer flakes. No existing theoretical framework or model, to the best of our knowledge, adequately explains these phenomena. Furthermore, using density functional theory (DFT) to calculate the local density of states (LDOS) in triangular MoS$_2$ flakes, we demonstrated how the balance between these two channels —metallic and semiconductor— can be finely controlled by adjusting the intensity of illumination with appropriately tuned wavelength. These theoretical insights guided our experiments, which confirmed that it is possible to precisely tune the interplay between the metallic and semiconducting phases.

These experimental results, in line with the theoretical framework, lead to an unprecedented level of control over the spin-pumping behaviour of the system, allowing us to amplify, diminish, switch on, or completely turn off the spin-pumping effect. We believe that this ability to finely control spintronic effects offers a significant breakthrough in the field of Hall effects and spintronics in general, with potential implications for advanced applications in spin-based devices.

## Results

To investigate the different contributions of the Hall effects to the spin pumping in molybdenum disulfide (MoS$_2$), we fabricated several YIG/MoS$_2$ heterostructures, as illustrated in Fig. 1. Monocrystalline yttrium iron garnet (YIG) thin films oriented along the (111) direction were sputtered on the top of a gadolinium gallium garnet (GGG) substrate to act as spin injectors into MoS$_2$[50,51]. The triangular-shaped MoS$_2$ monolayer flakes were synthesized by atmospheric pressure chemical vapor deposition (APCVD) on SiO$_2$/Si substrates[20,21]. As the growth of MoS$_2$ crystals starts at a nucleation point, the flake size can be controlled, ranging from 1 μm in lateral size to a continuous monolayer film by increasing the growth time. It is important to note that larger flakes and even films are formed by merging two or more flakes. The previously prepared and magnetically characterized YIG samples were then covered with different amounts and sizes of MoS$_2$ flakes via an etching-free transfer method[22].

To support the flake quality and the transfer method, Section S1 of the Supplementary Information shows and discusses the optical microscopy maps, Raman spectra, Raman map, photoluminescence spectroscopy, atomic force microscopy, and light absorption spectra of samples. The Methods and Experiments section describes sample preparation, Raman and photoluminescence spectroscopy, scanning electron microscopy, ferromagnetic resonance, spin pumping measurements, and density functional theory calculations.

The spin-to-charge interconversion can be analyzed through the thermodynamic Kelvin-Onsager reciprocal effects when the magnetization precession of the YIG injects a spin current into MoS$_2$. It is important to note that the YIG has very low Gilbert damping and, as a magnetic insulator, the angular momentum injection is expected to occur as a pure spin current, thus avoiding unexpected electrical effects that could lead to artefacts in the measurements[48,49]. Therefore, as proposed by Tserkovniak et al. [52,53] for SHE (which can be extended to OHE and VHE), the enhancement of the magnetic layer (FM) Gilbert damping when coupled to a metallic layer (M) is proportional to the Hall effects following the equation:

$$
\begin{aligned}
\alpha_{SP} &= \alpha_{FM/M} - \alpha_{FM} \\
&= \alpha_{YIG/MoS_2} - \alpha_{YIG} \\
&= \frac{g_L g^{\uparrow\downarrow}}{4\pi\mu} \left[ 1 + \frac{\sqrt{3} g^{\uparrow\downarrow}}{\sqrt{\varepsilon} S k_F^2 \tanh\left(\frac{L}{\lambda_{SD}}\right)} \right]^{-1} \\
&\approx \frac{g_L k_F^2}{(4\pi)^2 \mu} S \left[ 1 + \frac{\sqrt{3}}{4\pi\sqrt{\varepsilon} \tanh\left(\frac{L}{\lambda_{SD}}\right)} \right]^{-1}
\end{aligned}
\tag{1}
$$

where YIG is the FM layer and we are considering the MoS$_2$ (triangular flake or film) in the place of the metallic layer, so $\alpha_{YIG/MoS_2}$ and $\alpha_{YIG}$ are the Gilbert dampings for YIG/MoS$_2$ and YIG, respectively, $\varepsilon$ is the spin-flip probability at each scattering which is proportional to the spin-orbit coupling, $S$ is the interface YIG/MoS$_2$ area, $g_L$ is the g factor, $\mu$ is the total film magnetic moment in units of $\mu_B$, $g^{\uparrow\downarrow}$ is the interfacial mixing conductance, $k_F$ is the Fermi wave vector, $L$ is the YIG film thickness, and $\lambda_{SD}$ is the spin-diffusion length.

In Eq. 1, it is clear that spin pumping is influenced by the interface area ($S$) and the thickness ($L$) of the metallic film. The thickness dependence is related to the Hall effects in MoS$_2$, where spin pumping increases for smaller thicknesses and saturates for thicker films[54–57]. There's an ongoing debate about the mechanism of the spin-to-charge conversion phenomenon in semiconductor MoS$_2$. Experimental studies suggest the Rashba-Edelstein effect as the primary effect[31–35], while

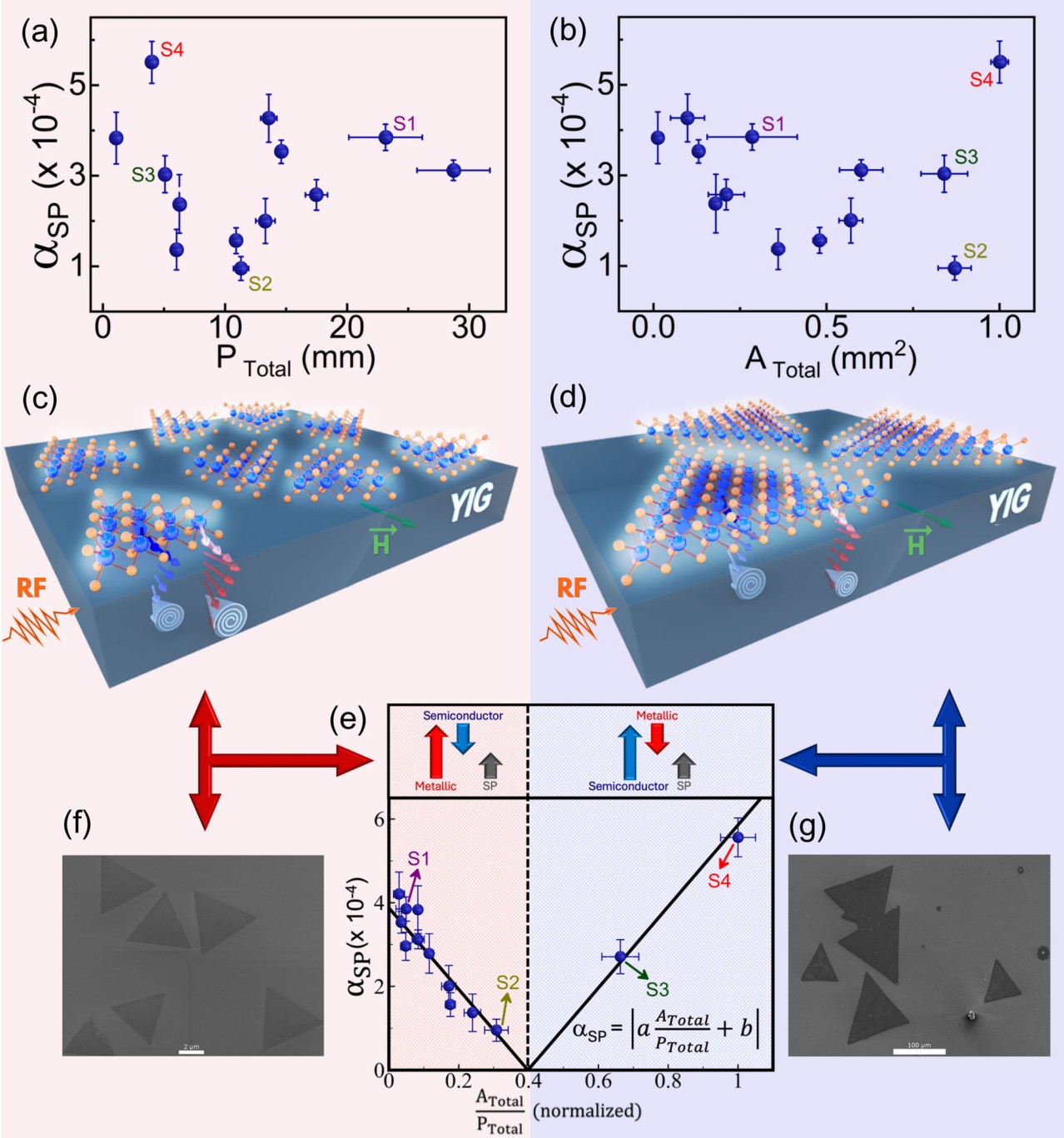

**Fig. 1 | Disentanglement of the spin-to-charge interconversion in MoS₂ mono-layer flakes.** Spin pumping ($\alpha_{SP}$) dependence as a function of the: (**a**) total edge length ($P_{Total}$), (**b**) total MoS₂ area ($A_{Total}$), and (**e**) total MoS₂ area divided by the total perimeter ($A_{Total}/P_{Total}$). The normalization on the (**e**) x-axis is based on the fact that the metallic edge states are proportional to the perimeter multiplied by the width of the metallic edge states, resulting in an adimensional unit. The

highlighted samples S1 to S4 were selected for studying the light influence in the SP. Illustrations of the YIG/MoS₂ heterostructures with small (red region) and larger flakes (blue region) are shown in (**c**) and (**d**), respectively. Representative scanning electron microscopy of MoS₂ flakes with small $A_{Total}/P_{Total}$ ratio is shown in (**f**) with a scale bar of 2 μm and with higher $A_{Total}/P_{Total}$ ratio in (**g**) with a scale bar of 100 μm.

theoretical works attribute it to Hall effects[58,59]. Both the REE and its Kelvin-Onsager reciprocal, steamed inverse REE (IREE) results from a broken inversion symmetry, which, for MoS₂, is expected only at the YIG/MoS₂ interface, making it independent of MoS₂ thickness. The Supplementary Information Section S2 details the thickness-independent behavior of spin pumping[51], leading to the conclusion that the semiconductor state dictates the main spin pumping behavior as expected for IREE.

On the other hand, the contribution of the MoS₂ metallic edge states to the spin pumping remains unexplored. It is worth noting that when MoS₂ is grown in a sulfurrich atmosphere, the MoS₂ triangular flakes terminate in a zig-zag-type edge[23–25]. Both theoretical and experimental studies state that the MoS₂ zig-zag termination produces metallic (conductive) edge-states[60–62].

To explore the influence of these edge states in the spin pumping, a set of samples varying the MoS₂ triangular size and the number of

flakes deposited onto the YIG film were measured by spin pumping. Henceforth, we define two important parameters: the total area of the $MoS_2$ flakes ($A_{Total}$) and the total perimeter of the $MoS_2$ flakes ($P_{Total}$). These parameters are calculated using the following equations:

$$A_{Total} = \sum_{i=1}^{n} a_i \qquad (2a)$$

$$P_{Total} = w \cdot \sum_{i=1}^{n} p_{i,} \qquad (2b)$$

The total area, $A_{Total}$, is the sum of the areas of each $MoS_2$ flake, $a_i$, covering the YIG film, where $n$ is the number of flakes. Similarly, $P_{Total}$ is the sum of the perimeters of the individual $MoS_2$ flakes, $p_i$; $w$, is a constant width of a few nanometers along the edges of the triangle, to provide a two-dimensional aspect to $P_{Total}$. To measure the area $a_i$ and perimeter $p_i$ of individual flakes, images obtained by optical microscopy were analyzed by the ImageJ software[63]. The $MoS_2$ merged edges and grain boundaries were not included in the total perimeter estimate. For more information on quantifying the area and perimeter of the samples, as well as their appearance, please refer to Section S1.8 of the SI.

The spin pumping dependence of $P_{Total}$ and $A_{Total}$ are shown in Fig. 1a, b, respectively. In previous discussions, it has been established that the semiconductor phase of $MoS_2$ exhibits IREE as the primary mechanism for spin pumping. Since this effect occurs at the interface, the spin pumping in these heterostructures is expected to be directly proportional to the YIG/$MoS_2$ interfacial area ($S$), as documented in refs. 31,33–37. If the spin pumping in the YIG/$MoS_2$ heterostructures were solely attributable to the semiconductor phase, one would anticipate a linear increase in $\alpha_{SP}$ with $A_{Total}$. However, as illustrated in Fig. 1b, it is evident that $\alpha_{SP}$ does not exhibit a linear relationship with $A_{Total}$, indicating that factors other than interfacial area need to be taken into account. This suggests that the metallic edge states could also play a role in the injection of spin current into the $MoS_2$ flakes.

Moreover, if only the metallic edge states, which are the areas along the sides of the triangles[26–30], contribute to spin pumping due to the inverse spin Hall effect (ISHE) as predicted by Eq. 1, for the case where a metallic layer is in direct contact with the FM, there should be a linear relationship between $\alpha_{SP}$ and the total edge $P_{Total}$. Figure 1a contradicts this expectation as there seems to be no correlation between the spin pumping and $P_{Total}$. Based on the analyses conducted so far, it appears that $\alpha_{SP}$ does not depend solely on either $A_{Total}$ or $P_{Total}$. This suggests that a more complex scenario needs to be considered to understand the origin of spin current injection into the $MoS_2$ flakes. To comprehend this point, both semiconductor area and metallic edge states have to be taken into account.

In order to differentiate between the contributions of the semiconductor area states and the metallic edge states, Fig. 1e illustrates the variation of $\alpha_{SP}$ as a function of the ratio of the total $MoS_2$ coverage area to the total $MoS_2$ perimeter ($A_{Total}/P_{Total}$). The graph displays a V-shaped curve with two distinct behaviors. The first, highlighted in light red, demonstrates a decrease in spin pumping as the $A_{Total}/P_{Total}$ ratio increases. Conversely, after a certain compensation point, where $\alpha_{SP}$ is extrapolated to zero (highlighted in light blue), the slope becomes positive, and $\alpha_{SP}$ increases as $A_{Total}/P_{Total}$ increases. Notably, the data can be fitted by the absolute value of a single linear function, as shown in Fig. 1e, where the solid line represents the experimental data fit by the equation displayed on the graph. Despite the complexity of the sample preparation, the entire dataset follows a linear behavior with minimal dispersion. These results suggest that both semiconductor area states and metallic edge states contribute to the injection of spin current into the $MoS_2$ triangular flake.

This V-shaped curve can be explained by a competitive interplay between the two channels contributing to the overall spin current injection. One channel is associated with the semiconductor area states, proportional to $A_{Total}$, and the second is associated with the metallic edge states, proportional to $P_{Total}$. To better understand those behaviors, it is helpful to visualize the diagram located above the graph in Fig. 1e. This diagram displays the red and blue arrows that represent the metallic edge and the semiconductor area channel contributions to the $\alpha_{SP}$, respectively. Meanwhile, the gray arrow indicates the overall measured spin pumping, which corresponds to the difference between these channels.

It is worth noting that when considering the samples with small $A_{Total}/P_{Total}$ ratio (light red region), they have a higher proportion of total edge ($P_{Total}$) compared to the area ($A_{Total}$) because they have smaller $MoS_2$ flakes. A representative scanning electron microscopy of the flakes of this region (light red) is shown in Fig. 1f with a 2 μm scale bar. To better visualize the largest metallic edge contribution in relationship to the semiconducting area in this region, a scheme of the spin pumping in these samples is depicted in Fig. 1c. The semiconductor contribution becomes more significant as the $A_{Total}/P_{Total}$ ratio increases since the samples in the light blue region have bigger flakes. An illustration showcasing the relationship between these two contributions in samples is depicted in Fig. 1d, and representative scanning electron microscopy of a bigger flake is displayed in Fig. 1g with a 100 μm scale bar.

To better understand this picture, first we focus on the light red region, where the $A_{Total}/P_{Total}$ ratio is small. Here, the contribution from the metallic edge state channels dominates, as represented by the red arrows in the diagram of Fig. 1e, leaving the semiconductor area state channels a secondary role. This area contribution has an opposite sign and is denoted by the smaller and opposite blue arrows in the diagram. Increasing the ratio, the metallic edge state channel's dominance remains, but its importance decreases. This means that the spin current injected by the semiconductor area states becomes more noticeable as the ratio increases. Considering the channels as having opposite polarity, a decrease in the overall spin pumping is observed, represented by the gray arrow in the diagram. This trend continues until, by extrapolation, both channels balance each other at the compensation point, where the value of $\alpha_{SP}$ would be zero, and therefore no spin current is injected into the $MoS_2$. Actually, so far, it is not clear why area and edge states have opposite contributions, but as we will discuss later, ab initio calculations will be useful to clarify this question.

Beyond the compensation point (light blue region), a further increase in the $A_{Total}/P_{Total}$ ratio leads to a switch in dominance. Therefore, the semiconductor area states, here represented by the largest blue arrows on the right side of the diagram (see Fig. 1e), dominates the spin current injection, while the metallic edge state, represented by the smaller and opposite red arrows plays a secondary role. A further increase in the $A_{Total}/P_{Total}$ ratio implies that the already dominant semiconductor area states become even stronger than the metallic edge states, leading to an overall increase in spin pumping. The V-shaped behavior observed for spin pumping ($\alpha_{SP}$) as a function of the $A_{Total}/P_{Total}$ ratio appears to be robust. In spin pumping studies, it is also common to consider the mixed spin conductance ($g_{\uparrow\downarrow}$) alongside $\alpha_{SP}$, as $g_{\uparrow\downarrow}$ depends on both $\alpha_{SP}$ and the magnetic properties of the ferromagnetic material. This makes it easier to compare across different ferromagnetic materials. In Supplementary Information Section S3, $g_{\uparrow\downarrow}$ is presented in more detail, along with its dependence on the $A_{Total}/P_{Total}$ ratio, which also exhibits the same V-shaped behavior.

The geometric properties of monolayer $MoS_2$ flakes have a significant impact on the competition between semiconductor and metallic edge states in spin injection to $MoS_2$. Each point in the graph in Fig. 1e corresponds to a different sample, so new samples are needed to better track the V-shaped curve in detail and uncover missing information about the origin of the spin injection, especially near the

compensation point and in the right region. However, producing new samples with a given $A_{Total}/P_{Total}$ in this curve is a challenging task.

To further investigate the V-shaped curve, we can adjust the balance between the metallic and semiconductor channels by exciting electrons from the valence to the conduction band. This can be done by illuminating the $MoS_2$ flakes with the appropriate light wavelength. By controlling the light intensity, we can regulate the number of electrons promoted to the conduction band, thus controlling the enhancement of the metallic contribution of the spin pumping and precisely tracking the V-shaped curve.

This approach is supported by density functional theory calculations of the electronic properties of $MoS_2$ flakes[64,65] (see Methods). To investigate the metallic behavior of $MoS_2$ triangular flakes, we analyzed atomic projected quantities, such as the LDOS and the real-space representation of the charge density.

The Fig. 2 panel A shows the LDOS for a $MoS_2$ triangular flake with a lateral distance of 57 nm and zigzag-terminated edges. We calculated the LDOS for two distinct regions: edge and area. The edge region is formed by the two outmost $MoS_2$ triangular flake atomic lines (green curve). Whereas, all other atoms form the area region (red curve). The shaded region corresponds to the $MoS_2$ monolayer bandgap (fully periodic), which corresponds to a value of 1.7 eV. This bandgap value is in good agreement with the literature[66] (see Supplementary Information Section S4). As shown in panel A, the area LDOS (red curve) is very small within the monolayer bandgap (shaded region). In contrast, the edge LDOS (green curve) exhibits a high density of states within the bandgap, indicating the presence of metallic edge states in the $MoS_2$ triangular flake with zigzag termination, as previously discussed. A deeper understanding of the charge spatial distribution can be obtained by examining the partial charge density (PARCHG) at selected energy levels. The vertical blue lines labeled (i), (ii), (iii), and (iv) in

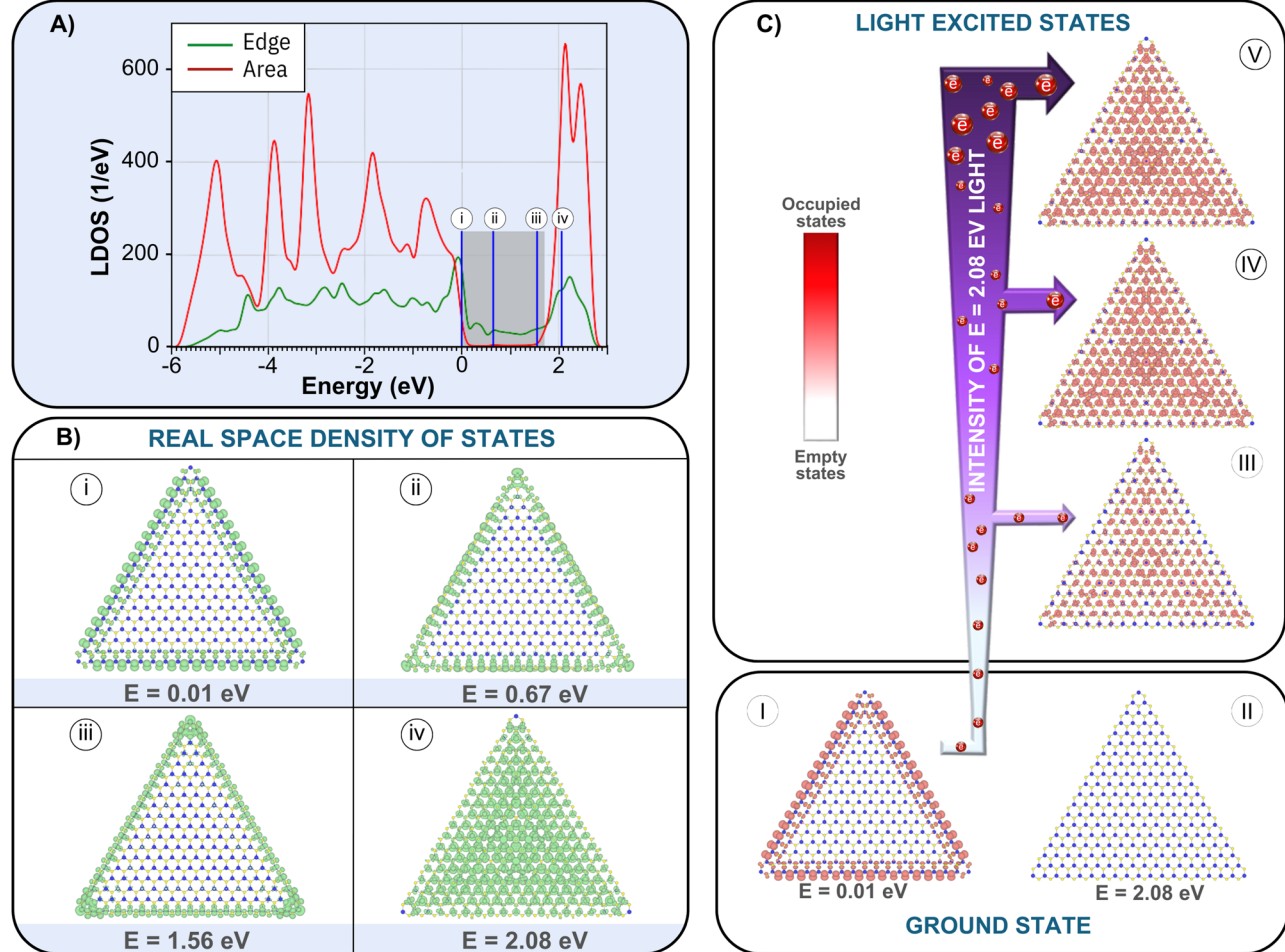

**Fig. 2 | Area and edge MoS₂ electronic states.** In (**A**), the graph depicts the local density of states (LDOS) as a function of the energy for a $MoS_2$ triangular flake with zig-zag termination and a lateral distance of 57.3 nm. The green curve represents the edge states LDOS, defined as the two outermost atomic lines along the flake's edge. The red curve is the area states LDOS, which is defined as all other atomic sites on the flake. The grey region highlights the full $MoS_2$ monolayer (fully periodic) bandgap energy region. The vertical blue lines labeled (i), (ii), (iii), and (iv) correspond to selected energies, namely, 0.00, 0.45, 1.67, and 2.08 eV, respectively. **B** shows the $MoS_2$ triangular flake real space representation of the partial density of states for each energy highlighted in panel (A). The isosurface value is 0.005 electrons/Å³. **C** illustrates the process of electronic excitation in the triangular $MoS_2$ flake when exposed to 2.08 eV light, which promotes electrons from below the Fermi level, as depicted in panel B (i), to the unoccupied states above the bandgap (spatial distribution shown in panel B (iv)). Initially, the system is in its ground state,

with electrons occupying states below the Fermi level, as represented by flake (I), leaving the states above the bandgap entirely unoccupied, as shown in flake (II). Upon illumination, electrons are excited from the Fermi level to higher-energy states, filling previously unoccupied states. As the intensity of the light increases, electrons preferentially populate states with the highest LDOS, as illustrated by flake (III). With further increases in light intensity, more electrons are excited from the Fermi level, progressively occupying states that are less probable, as reflected by the increasing red contrast in the flakes (II-V). At maximum light intensity, all available states become occupied, rendering the flake fully metallic, as demonstrated by flake (V). Notably, the spatial distribution of LDOS in the fully illuminated flake (V) mirrors that in panel B (iv), with the key difference being that the states in flake (V) are fully occupied, whereas those in panel B (iv) remain unoccupied. The color scale ranges from white, representing no empty states, to red, representing the occupied states (i.e., the metallic states), as shown on the scale bar.

panel A, correspond to energies of 0.00, 0.45, 1.67, and 2.08 eV, respectively.

The charge distributions at these selected energies are further analyzed in Fig. 2 panel B, from (i) to (iv). These panels depict the real-space representation of the PARCHG for the $MoS_2$ triangular flake at each corresponding energy. Panel B (i), shows the spatial charge density distribution at the Fermi level (E = 0.00 eV), which represents the ground state of the system. This result reveals that the most populated sites are located at the edges of the flake, which confirms its metallic behavior. The flakes in panel B (ii) and (iii) show the spatial distribution of the LDOS at excited energy states within the system's bandgap. As previously discussed, the LDOS within the bandgap is negligible (as indicated by the red curve in the shaded region of panel A). The edge states dominate, so the spatial distribution of the LDOS at these energies does not change significantly. In contrast, panel B (iv) shows the LDOS at an excited energy state beyond the bandgap. Unlike the previous PARCHG, the LDOS is more evenly distributed across both the surface area and the edges of the flake.

Figure 2 panel C illustrates the process of electronic excitation in the flake when exposed to 2.08 eV light. This light promotes electrons to unoccupied states above the bandgap, as shown in panel C (I) and (II). The red sites represent occupied states, while the white sites represent vacant ones. Initially, the system is in its ground state, with electrons occupying states below the Fermi level, as depicted panel C (I) (or equivalently panel B, (i)), leaving all other states unoccupied. When the flake is illuminated, electrons are excited to higher energy states, filling previously unoccupied states. As the intensity of the 2.08 eV light increases, electrons are first excited to the most probable states—those crystalline sites with the highest LDOS—as shown in panel C (III). With further increases in light intensity, more electrons near the Fermi level are excited to unoccupied states above the bandgap, as depicted in panel C (IV). These electrons gradually occupy less probable sites, i.e., those with lower LDOS. This process is represented by the increasing red contrast in the flakes in panel C (II) to (V), reflecting the growing number of electrons occupying these states. With a further increase in light intensity, the red contrast intensifies (panel C (IV)), indicating a greater number of sites being occupied by electrons, effectively rendering more metallic sites. Eventually, at a given light intensity, all states are occupied, and the flake becomes fully metallic, as represented by the red triangle in panel C (V).

When analyzing the effect of the 2.08 eV light illumination (see Fig. 2 panel C (II-V)), it is evident that electronic excitation, leading to the creation of light-induced metallic sites, initiates from the central region of the flakes and spreads towards the edges as the light intensity increases. Notably, the spatial distribution of LDOS in the fully illuminated flake (panel C (V)) resembles that of Panel B (iv), except that initially the states (panel B (v)) are occupied, while those at the second (Panel B, (iv)) remain unoccupied.

Our results indicate that in the absence of any significant doping, meaning the Fermi level is at zero energy, our samples will show a metallic character that is controlled by the edge states.

According to this theoretical approach, we can analyze the V-curve by focusing on the compensation point. This involves shining the appropriate light wavelength on the $MoS_2$ flakes, as shown in Fig. 2. As a result, one can expect the semiconductor's contribution to spin pumping to decrease while the metallic contribution increases. By adjusting the light intensity, it is possible to control the number of electrons moving to the conduction band, which in turn regulates the enhancement of the metallic contribution to spin pumping. Therefore, the V-curve can be accurately tracked by controlling the light intensity and fine-tuning the balance between the semiconducting area and metallic edge states.

To study the effects of light on the spin current generation, we have performed spin pumping measurements in the presence of light with different wavelengths to populate the conduction band of $MoS_2$,

according to the illustration in Fig. 3a. The DFT bandgap of $MoS_2$ was calculated to be 1.7 eV. However, it is well known that DFT tends to underestimate the actual bandgap (see Supplementary Information Section S4). Therefore, in the experiments, we chose to use violet light with a wavelength of 405 nm (3.03 eV) which is expected to be above the electronic bandgap. We selected four samples with different $A_{Total}/P_{Total}$ ratios to cover the two regimes. These four samples are labeled S1, S2, S3 and S4 in Fig. 1. As can be seen in this figure, through the V-shape, the samples S1 to S4 are in order from left to right, i.e. for smaller to larger flakes. Remembering that this is not the case for either $A_{Total}$ (Fig. 1b) or $P_{Total}$ (Fig. 1a). The spin pumping variation of these four samples and a bare YIG sample is shown in Fig. 3(b), where the samples clearly exhibit different behaviors. To provide a more detailed explanation of each sample, the results of S1-S4 are shown in Fig. 3c through f, respectively. The diagram at the top of each figure uses the same notation as before: the red arrows indicate the metallic edge, while the blue arrows represent the 2D semiconductor contributions in the opposite direction. The gray arrows represent the total spin pumping measured, which is the difference between the metallic edges and 2D semiconductor contributions. The purple arrows represent the light-driven contributions.

The sample S1 is represented by purple squares in Fig. 3c. It has the smallest $A_{Total}/P_{Total}$ ratio, far before the compensation point, lying in the light red region (see Fig. 1e). Here, the contribution of metallic edge state channels to spin pumping is more significant than that of semiconductor area states. On the top of this figure is a diagram that illustrates the contribution to overall spin pumping (the measured $\alpha_{SP}$). This sample presents a dominant metallic edge state (red arrwos) while the smaller 2D semiconductor contribution is in the opposite direction. By illuminating the S1 sample with violet light, electrons within the semiconductor regions are excited from the valence to conduction bands, as indicated by the violet arrows. This photo-excitation results in an increase in the density of conductive carriers in the semiconductor areas, thereby enhancing the metallic contribution to the $\alpha_{SP}$, which continues to rise with intensified light exposure.

Sample S2 is represented by yellow pentagons in Fig. 3d; it has the same behavior as S1. In both samples, the spin pumping is already dominated by the metallic edge state (red arrows in the diagram), which has the same contribution intensified by the light (violet arrows in the diagram). Therefore, the enhancement of spin injection (gray arrow in diagram) is achieved by increasing the photon flux.

It is worth noting that when $MoS_2$ is illuminated, there is a significant increase in spin injection from YIG to $MoS_2$ in both S1 and S2 samples. For sample S1 the spin injection doubles from $\alpha_{SP} = 3.2 \times 10^{-4}$ to $\alpha_{SP} = 6.5 \times 10^{-4}$, when illuminated. For comparison, this value is in the same order as the YIG/Pt system, among the highest values reported so far[50,51]. These findings are particularly impressive given the low coverage of TMD ($A_{Total}$) in the S1 sample, which only covers about 28% of the YIG's total area (Fig. 1b). Additionally, one can notice that the light-driven spin pumping enhancement in sample S1 (Fig. 3c) is higher than in sample S2 (Fig. 3d). This is possibly due to the number of available states depending on the average size of flakes. The overall $MoS_2$ covered area of sample S1 is much smaller than that of sample S2, therefore, the light intensity required to achieve the flakes fully metallic (as exemplified in Fig. 2 panel C (V)) could be smaller for smaller flakes.

In sample S4, represented by red hexagons in Fig. 3e, the $A_{Total}/P_{Total}$ ratio is the highest and far beyond the compensation point lying in the light blue region (see Fig. 1e). As illustrated in the diagram the 2D semiconductor state represented by the blue arrows dominates the spin pumping. The metallic edge state contribution to the spin pumping in this region is smaller, with an opposite signal, as illustrated by the smaller switched red arrows. As the light enhances the metallic states, this contribution (represented by the violet arrows) will be additive to the metallic edge contribution. Hence, as the overall spin

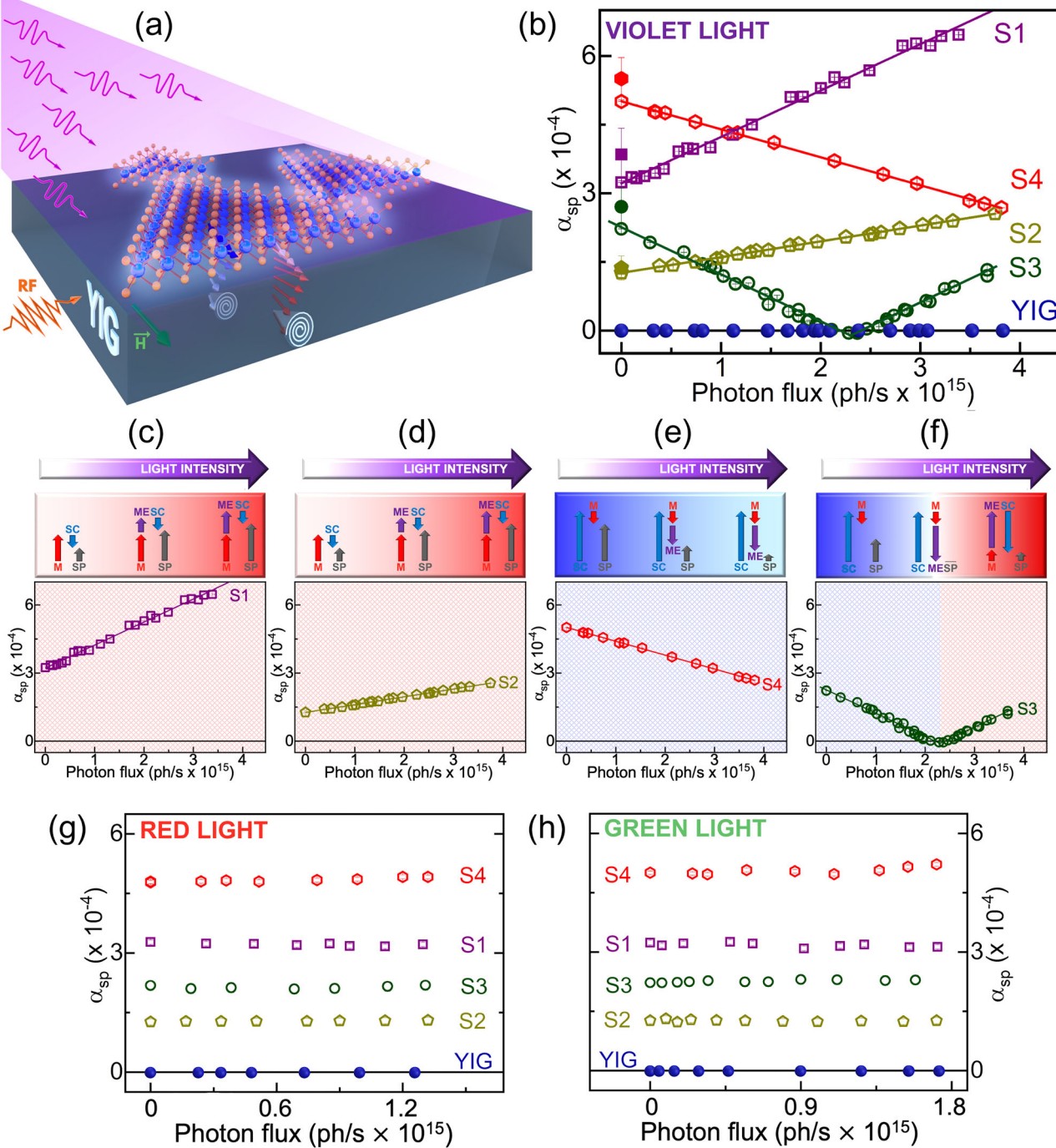

**Fig. 3 | Light-driven spin-to-charge conversion.** In (**a**) is presented a schematic representation of the light-induced spin pumping. The spin pumping ($\alpha_{SP}$) as a function of photon flux (light intensity) is shown in (**b**) to (**f**) for violet light (405 nm), (**g**) for red light (650 nm), and (**h**) for green light (532 nm). In the diagrams (**c**) to (**f**) the overall spin pumping (SP) and the individual contributions from the metallic (M), semiconducting (SC) and metallic excited (ME) states are represented by gray, red, blue and purple arrows, respectively. Samples S1 to S4 are represented by purple squares, yellow pentagons, green circles, and red hexagons, respectively. The solid symbols represent the data recorded on the coplanar waveguide without light incidence, and the open symbols represent the data recorded in the cavity with light incidence. The solid blue circles represent the bare YIG.

pumping measured is the difference between metallic and semiconductor channels, the overall spin pumping (gray arrows) will experience a decrease with the increasing photon flux.

The sample S3 represented by green circles in Fig. 3f is noteworthy. The nonilluminated sample is in the same region as sample S4. However, the $A_{Total}$ to $P_{Total}$ ratio is close to the compensation point, where $\alpha_{SP}$ is zero. As deciped in the diagram, at lower photon flux, the sample S3 behaves similarly to the S4 sample. This means that the light

enhances the metallic contribution, which has an opposite signal regarding the semiconductor contribution, reducing the total $\alpha_{SP}$, as represented by the leftmost set of arrows on the diagram. This trend continues as the photon flux increases until the contribution of the metallic states reaches the same intensity as the semiconductor. At this point, both contributions cancel each other, and the overall spin pumping is zero. The central set of arrows in the diagram represents this compensation point. As depicted by the rightmost set of arrows,

the system switches with a further increase in photon flux. The metallic states become the major contributor to $\alpha_{SP}$, in detriment to the 2D semiconductor contribution. From this point, increasing the light intensity will promote the already dominant contribution, the metallic state. At this point, sample S3 will behave like the S1 and S2 samples, meaning that the overall $\alpha_{SP}$ will increase with the photon flux. In fact, with higher photon flux, the sample S4 is expected to behave similarly to the S3.

For comparison, Fig. 3b shows the $\alpha_{SP}$ measure for all samples and the bare YIG (as reference sample) as a function of the violet light (405 nm and 3.03 eV) intensity while keeping the symbols and the color notation of the samples.

A recent theoretical work reported by Habara and Wakabayashi[67] has shown that in a monolayer of metallic NbSe$_2$, a TMD with strong spin-orbit coupling field, this field can act as an effective Zeeman field, leading to unconventional topological spin properties. The authors predict that a pure spin Hall current can be generated by light irradiation due to the topological nature of monolayer NbSe$_2$ and its finite spin Berry curvature. As the same ingredients are also present, a similar effect can likely occur in MoS$_2$. Since NbSe$_2$ is metallic, one can conjecture that illumination with a suitable light (violet here) can enforce the already existent metallic edge states spin injection from YIG to MoS$_2$ flakes. This explanation might be the microscopy origin of enhancing the metallic state contribution driven by light. In addition, our results also support the predictions made by the authors, i.e., the light-driven spin-polarized current in TMDs.

Further tests were performed with the samples under different conditions to ensure that the observed light-induced effects were done by promoting electrons to the conduction band and not due to spurious effects or any measurement artifact. To verify that the promotion of electrons was responsible for the observed effects, $\alpha_{SP}$ was measured as a function of photon flux for the four samples and a bare YIG sample for distinct wavelengths. Figure 3g, h show the spin pumping with the samples excited by red light (1.91 eV and 650 nm) and green light (2.33 eV and 532 nm), respectively. By using these two wavelengths, no change in $\alpha_{SP}$ was observed. Therefore, for these energies the electrons cannot be excited into the conduction band, which does not change the measured total spin pumping. It is important to note that although the photon energy of the green light (2.33 eV) is above the underestimated bandgap calculated by DFT, as discussed earlier and in the SI, it is probably still smaller than the actual bandgap. This is in agreement with recent experiments that have measured for the MoS$_2$ monolayer an electronic bandgap as 2.40 eV[68]. YIG sample represented by the blue circles in Fig. 3 was measured under the same conditions as the previous samples, and no change in the FMR linewidth was found, regardless of whether or not red, green or violet lights were used. This confirms that the observed behavior is due to MoS$_2$.

Recent studies have shown that ultraviolet (UV) light can irreversibly alter the properties of MoS$_2$. For instance, UV exposure can modify the interaction between MoS$_2$ and oxygen, promoting the formation of oxides on the MoS$_2$ surface[69,70]. Furthermore, UV light can introduce defect states and modify the surface bonding of MoS$_2$, leading to changes in its band structure[71]. Although these irreversible effects have been observed with UV light, which has a higher photon energy, it is crucial to verify that the violet light used in this study does not induce similar alterations or damage to MoS$_2$. To ensure that violet light did not damage the MoS$_2$ flakes, we performed FMR measurements before and after all experiments with varying incident light power. As shown in Supplementary Information Section S5 the FMR spectra and $\alpha_{SP}$ remained unchanged before and after the experiments with light. This indicates that the light did not damage the MoS$_2$ monolayer and that the experimental procedure is fully reversible.

All the narratives until now need further explanation about the origin of spin pumping in a semiconductor material. MoS$_2$ has a gap of approximately 2 eV, as discussed in detail throughout the manuscript. Therefore, it is unclear how the excitation employed in the FMR essays, which is in microwave order, can inject the angular momentum from YIG to MoS$_2$.

Considerable research has been conducted in the field of energy-efficient magnetization manipulation. To minimize energy consumption in magnetization switching, researchers have focused on using light-assisted spin-orbit torque to control magnetization and spin current generation in TMDs[72–74]. Moreover, recent studies have demonstrated the possibility of electrically controlling the modulation of circular polarization and spin injection through magnetization dynamics. This breakthrough is a significant step towards the development of next-generation information and communication technology[75–77]. However, energy-efficient opto-spintronic devices are still far from being achieved.

This work represents a significant advance in the understanding of the physical nature of spin-charge interconversion in transition metal dichalcogenides. Our results show distinct contributions from 2D semiconductor area and metallic edge states to spin current injection in MoS$_2$ and the role of light-excited states in these contributions. Beyond the fundamental findings, our results allow us unprecedented control over spin current injection. By exploiting the light intensity, the spin pumping can be finely tuned by promoting light-excited states where the spin currents can be amplified, attenuated, or even switched on and off. Therefore, these results mark a significant milestone in opto-spintronics by demonstrating the feasibility of controlling spin currents via light excitation at room temperature.

## Discussion

We studied the efficiency of light-driven spin current injection in triangular MoS$_2$ monolayer flakes from YIG thought spin pumping measurements. Alongside the detailed fabrication and comprehensive characterisation, the main experimental results reported are: The identification of two distinct channels contributing to spin pumping, one originating from metallic edge states and the other from semiconductor area states. On this regard, we can conjecture that each may have a different microscopic origin. The semiconductor phase, as demonstrated in this work, is driven by the inverse Rashba Edelstein effect, while the metallic phase likely arises from the spin Hall effect.

This dual contribution opens up two promising directions for these systems, either in device applications or as fundamental prototypes for deeper studies in spintronics.

Our experimental evidence clearly demonstrates that these two channels operate concurrently. While one channel transfers angular momentum from the ferromagnetic layer (YIG) to the TMD (MoS$_2$, in our particular case), the other channel either returns angular momentum from the TMD to the YIG or inject angular momentum with an opossite sign. This competition gives rise to a compensation point where these spin-pumping contributions cancel each other out. The existence of various channels contributing to spin-to-charge interconversion is well-established in the literature, with these mechanisms typically realized through the use of multiple layers, either within the same material or across different materials. For instance, Pt/FM/W trilayers are often employed to enhance terahertz emission[78,79]. While Pt and W exhibit opposite spin Hall angles, their placement relative to the FM layer leads to a geometry in which the spin currents from both materials combine to enhance the emission. In contrast, Pt/FM/Pt multilayers have recently been used to investigate the orbital Hall effect in adjacent materials. In this case, the spin current injected by the bottom Pt layer is either partially or fully canceled by the contribution of the top Pt layer. This competition between the two Pt layers facilitates the study of the adjacent material[80,81]. Despite the distinct mechanisms of spin current injection, which in the case of Pt and W is the spin Hall effect, an analogy can be drawn to better

understand the interaction between the two spin current channels in $MoS_2$, the metallic edge and semiconductor area states.

It is noteworthy that the zigzag metallic edge states studied here may not be the only additional channel capable of spin-to-charge interconversion in TMDs. Recently, the existence of one-dimensional metallic states in $1H\text{-}MoS_2$ grain boundaries has been shown[82]. However, this particular configuration, called mirror twin boundaries, only occurs when the two adjacent crystals are exactly 60° rotated. In our case, the relative crystal orientation of the adjoining flakes was not controlled, so the formation of this specific case is unlikely. In addition, no experimental evidence for boundary influence on spin pumping has been found. Nevertheless, the existence of different one-dimensional metallic channels in TMDs opens the possibility of investigating new channels of spin-to-charge interconversion.

Moreover, we have shown experimentally that by shedding the system with light with suitable energy (wavelength) and adjusting its intensity, we can modulate the balance between metallic and semiconductor channels. This allows the fine control of the spintronic manifestation in this system, either amplifying, diminishing, switching them on, or even completely turning off the spin current injection. Our experimental light-driven disentanglement results between the metallic edge and semiconductor area phases were studied by the local density of state of the $MoS_2$ flake calculations and its real-space representation of the partial charge density (which depicted the spatial distribution of the LDOS), comprehensively conducted by density function theory calculations. These analyses helped to verify the concomitance of the states of the metallic edge and the semiconductor area and how the intensity of the light (with the right energy) can balance each other. In principle, a further spin pumping calculation could be achived by including many body interactions and linear response theory in a combination with DFT and effective Hamiltonians.

It is crucial to note that although we can offer some plausible insights into the underlying mechanisms, no existing model or theory, to the best of our knowledge, adequately explains this phenomenon. We believe that a study presenting such clear, reproducible, and highly unexpected results—results that challenge conventional thinking and stimulate new questions for the scientific community. Additionally, the ability to control the spintronic properties of this system -amplifying, attenuating, switching on, or completely turning off spin pumping- simply by modulating the intensity of incident light represents a breakthrough innovation. This capability not only introduces a previously unreported mechanism in the field but also opens the door to a wide array of energy-efficient optoelectronic devices.

The observed light-driven modulation of spintronic effects in $MoS_2$ likely extends across the broader family of transition metal dichalcogenides, including other 1H phase compounds (e.g., $MoSe_2$, $MoTe_2$, $WS_2$) and possibly the T and T' phases, some of which may exhibit topological insulating states. This generality suggests a significant foundation for further research, potentially expanding control over spintronic properties in two-dimensional materials. As detailed in recent literature[83], several TMD compounds may present similar behaviors, although some are still theoretical predictions. By broadening the applicability of our findings, our study paves the way for a new avenue in spintronic research within TMDs.

## Methods

### Yttrium iron garnet growth
Thin films of yttrium iron garnet (YIG) were grown by magnetron sputtering on monocrystalline gadolinium gallium garnet (GGG) substrates oriented along the (111) direction. The sputtering chamber pressure was $9.0 \times 10^{-8}$ Torr, and the working argon pressure and flow were 10 mTorr and 15 sccm, respectively. We used an RF power of 75 W. After deposition, an ex situ annealing was performed with oxygen flow. The films had a thickness of 50 nm and a square-shaped pattern of $1 \times 1$ mm², which were achieved through photolithography using a laser writer model μPG101 with 3 μm resolution and AR-P3510 photoresist[50].

### $MoS_2$ monolayer growth and transfer
Triangular flakes of $MoS_2$ monolayers have been synthesized by atmospheric pressure chemical vapor deposition (APCVD) on $SiO_2/Si$ substrates. Flakes with different edge sizes were obtained by varying the growth conditions, ranging from 1 μm to a continuous film. The growth of the $MoS_2$ crystal starts from a nucleation point and is obtained by sulphurisation of molybdenum oxide. By increasing the growth time, the size of the triangular flakes continues to grow and eventually coalesce to form a uniform continuous $MoS_2$ film. A given number of $MoS_2$ triangles with a chosen edge size were individually transferred from the Si substrate to the YIG using a simple etch-free transfer method. Finally, several samples of YIG with different areas of $MoS_2$ coverage were prepared following the procedure described in refs. [20–22].

### Surface and edge analysis
All samples were investigated by optical microscopy to evaluate the covered area and the total length of the edges of the $MoS_2$ triangles. Images were analyzed using the ImageJ software[63].

### $MoS_2$ few-layer exfoliation
A few layers of $MoS_2$ were deposited on YIG films using an automated technique based on mechanical abrasion. We used a soft polymer (polydimethylsiloxane, PDMS) to cover a tip coupled to a computer numerical control (CNC). This tip is then pressed into the $MoS_2$ powder precursor, and the CNC acts as an XY writing pad, performing mechanical abrasion and leaving TMD along the way, providing the deposition. The CNC is coupled to a piezoelectric sensor that controls the z coordinate to achieve excellent uniformity. This technique allows the deposition of TMD thin films on large substrates without damaging the substrate surface. Film properties are controlled through the parameters of the deposition system. The thickness of the film is mainly influenced by the number of exfoliations of the material on the substrate surface. Consequently, this approach made it possible to obtain a certain thickness by adjusting the number of exfoliations[51].

### Raman and photoluminescence spectroscopy
Raman and photoluminescence maps and spectra were obtained using a micro-Raman spectrometer (NT-MDT, NTEGRA SPECTRA) in a backscattering configuration equipped with a solid-state laser (473 nm). We performed the experiments using a 100× objective and an incident laser power of 0.2 mW. The Raman and PL mapping images were collected using a $10 \times 10$ μm² piezoelectric stage.

### Scanning electron microscopy
The scanning transmission electron microscopy (STEM) imaging was performed at an acceleration voltage of 3.0 kV on a Jeol 7100FT microscope.

### Ferromagnetic resonance and spin pumping
Ferromagnetic resonance (FMR) was performed on all samples before and after $MoS_2$ transfer. FMR measurements were performed using a broadband coplanar waveguide from 3 to 14 GHz with AC magnetic field modulation (0.5 Oe and 45 kHz) for lock-in detection. A fixed-frequency cavity configuration (9.8 GHz) was used with the same modulation techniques for light-excited FMR measurements[50,51]. More details on the FMR and spin pumping analysis can be found in the Supplementary Information Section S3.

### Density functional theory calculation
We used the plane-wave-based code Vienna ab-initio simulation package (VASP)[84,85]. With the generalized gradient approximation

(GGA)[86,87] to treat the exchange and correlation potential. The ionic cores were described using scalar relativistic projected augmented wave (PAW) potentials[88]. A plane-wave-expansion cutoff of 400 eV was adopted with a 0.01 eV/Å force criterion for structural optimization. The MoS₂ flakes were constructed with a triangular geometry and a zigzag edge termination, to reproduce the experimental setup. Different flake sizes were explored and we are showing results for the largest, with a lateral size of 57.3 Å and 604 atoms in the supercell. Since VASP uses periodic boundary conditions, we included a vacuum layer in all directions to ensure a minimum of 15 Å distance between an atom and its image.

## Supplementary information

Optical images of the Si/SiO₂/MoS₂ and GGG/YIG/MoS₂ samples. Detailed Raman, photoluminescence, and light absorption spectroscopy. Raman frequency maps and atomic force microscopy of single MoS₂ flake. Details of the ferromagnetic resonance and spin pumping measurements. A discussion about the bandgap in the MoS₂, and the Rashba-Edelstein Effect in the semiconductor states of the MoS₂.

## Data availability

The source data is available at https://doi.org/10.5281/zenodo.14976824. All relevant data are available in the main text, supporting information, and source data, and can be obtained from the authors upon request. Source data are provided with this paper.

## Code availability

The code is available at https://doi.org/10.5281/zenodo.14976862.

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

## Acknowledgements

The authors thank the Conselho Nacional de Desenvolvimento Cient´ıfico e Tecnol´ogico (CNPq), Fundac¸˜ao de Amparo `a Pesquisa do Estado do Rio de Janeiro (FAPERJ), the INCT of Spintronics and Advanced Magnetic Nanostructures (INCT-SpinNanoMag, 406836/2022-1), the Financiadora de Estudos e Projetos (FINEP) for financial support, and INCT Materials Informatics. The authors acknowledge the financial support from DPI/BCE/UnB under the Call for Proposals No. 001/2025 DPI/BCE/UnB. The authors thank LABNANO/CBPF for the lithography facilities. The authors thank Roberto Bechara, Antonio Costa, Bruno Pimentel, and Igor Evangelista for helpful discussions. R.T.V. is thankful for Grant No. E-26/201.677/2021 from FAPERJ/Brazil. M.C. acknowledges the financial support of CNPq (Grant No. 317320/2021-1) and FAPERJ/Brazil (Grant No. E26/200.240/2023). J.F.F. and J.F.R.M. acknowledge financial support from Fundac¸˜ao de Apoio `a Pesquisa do Distrito Federal (FAPDF, Grant No. 193.00001823/2022-10 and 00193-00002418/2023-08). V.C. acknowledges CNPq (Grant No. 311863/2021-3), FAPERJ/Brazil (Grant No. E-26/204.493/2024, E-26/201.394/2021, E-26/210.539/2019), and US Air Force (Grant No. FA9550-23-10329). F.G. acknowledges CNPq (Grant No. 307624/2022-6), FAPERJ/Brazil (Grant No. E26/200.995/2022).

## Author contributions

R.T.V. contributed to the conceptualization, investigation, project administration, supervision, and writing the original draft. J.F.R.M. and S.H.S. contributed to the investigation. M.C. contributed to the theoretical calculations. J.F.F., V.C., L.C.S., and F.G., contributed to conceptualization, project administration, and supervision. All authors contributed to the formal analysis, review, and editing of the manuscript.

## Competing interests

The authors declare no competing interest.
