## [Transparent Peer Review file · Nature Communications]

Light-driven disentanglement between metallic edge and semiconductor area states in MoS₂ monolayer flakes

Corresponding Author: Professor Flavio Garcia

Version 0:

Reviewer comments:

Reviewer #1

(Remarks to the Author)
See attached

Reviewer #2

(Remarks to the Author)

In this study, the authors examined the efficiency of spin current injection in MoS₂ monolayer flakes on YIG substrates through spin pumping measurements. They explored how the Gilbert damping factor α changes with the $A_{\text{Total}}/P_{\text{Total}}$ ratio by varying the size of the MoS₂ monolayer flakes. Their findings suggest that both the 2D semiconductor properties and the metallic edge states of MoS₂ contribute to the spin injection process. They also show that light-driven spin pumping can enhance, reduce, or even completely suppress spin injection in the YIG/MoS₂ heterostructures. The findings, while potentially valuable, do not present a breakthrough or novel insight that significantly advances the field of spintronics or 2D materials science. One key concern is the unclear reasoning behind the use of the $A_{\text{Total}}/P_{\text{Total}}$ ratio to explain the behavior of the Gilbert damping factor. The relationship between these parameters and the spin dynamics in the system is not sufficiently justified or explained, leaving readers with unanswered questions about the underlying mechanisms. Even after revisions to address the issues, the work may still not meet the high bar set by Nature Communications. The manuscript may be better suited for publication in other materials science journal after revision.

Reviewer #3

(Remarks to the Author)
Comments:

Overall, this is an interesting work. The goal of this work was to disentangle effects of the metallic edge and the semiconducting states on the spin jumping behaviours. As promising as it is, I suggest a major revision and hope that the following concerns could be carefully addressed before I can decide whether or not recommend its publications.

1. Please provide the values that are used for each term in Eq. 1 to obtain Fig. 1a and b, and cite references where they were originally mentioned if necessary.
2. The scale bar in Fig. S4 is missing. Please add it.
3. If the $g\uparrow\downarrow \approx (SkF^2)/(4\pi)$ as written in p. 5, and there is another S term in the later bracket of Eq.1, the α_{SP} seems not proportional to the S (i.e., the YIG/MoS₂ contact area). Please clarify why a linear correlation can be expected. Also, it seems that an equation (or even a theoretical model) to express the relation between α_{SP} and the Perimeter term was not provided. Therefore, the referee felt confused while reading the first paragraph in p. 6. Please clarify this.
4. How did the authors estimate the total perimeter of MoS₂ to obtain Fig.1a. As mentioned in Supplementary Information, the coverage of CVD-grown MoS₂ looks tunable. The referee wonders what kind of sample was used, i.e., flakes or films, for Fig. 1a. Did authors include the merged edge in the estimation? If the MoS₂ sample was a continuous film, how did the authors distinguish where the edges exactly are given that the contrast of merged boundaries should be extremely weak under OM.

5. Please explain what those unspecified data points are in Fig. 1e. it would be helpful if the authors specify and provide the appearance of each sample used.
6. In Fig. 2c-f, the figure captions for "ME", "SP", "SC", and "M" are missing.
7. In p. 6, the authors stated as follows, "When electrons are excited from the valence to conduction bands by violet light, the metallic contribution is amplified,...". This might not be reasonable since there is no band gap for metallic materials. Through the entire Fig. 2, the influence of light shining on the MoS₂/YIG samples were not clear. The referee's inference is that the light affects both the semiconducting and metal states, and the photon flux also promote the spin injection from the YIG to the MoS₂. However, if the light enhanced the metal state in S1 to S4, why the enhancement improved α SP for S1&2 while it decreased the α SP value for S3&4. Would that be possible for the intensified light to only enhance the semiconducting states for all the four samples? Please verify the mechanisms, supported with theoretical model in the main text.
8. Following Q8, it would be helpful if the explanation can be supported with band diagrams.
9. It has been well known that monolayer MoS₂ has a direct band gap 1.8~1.9 eV. The authors mentioned a different value of 2 eV, which was even believed an underestimated value. The referee wonders based on what the authors think the monolayer MoS₂ used in this work had a larger band gap. In addition, literatures also have shown that a red-light laser (with a wavelength ~633 nm) is resonant to the band gap of monolayer MoS₂ and enough to excite electrons, (which is why we can see the A-exciton peak in photoluminescence). The origin that accounts for the observation regarding Fig. 2g might need to be reconsidered.
10. In p. 11, the full name of SOT should be provided.

Reviewer #4

(Remarks to the Author)

Version 2:

Reviewer comments:

Reviewer #1

(Remarks to the Author)

The revised manuscript by Rodrigo et al. have tried their best to address the comments by all the reviewers. They have added supporting measurements and experimental details. They have developed theoretical background which tries to explain the influence of light on enhancing the metallic phase in their system at the expense of the semiconductor phase. They have modified the manuscript to highlight the novelty in a better way, which is the modulation of the balance between metallic and semiconductor channels using lights of suitable wavelength. This could help control the spin current injection in spintronic devices. I believe the manuscript has improved quite a lot, and many of the doubts have been cleared. However, there are still a few unanswered questions, especially regarding the mechanism. Their statement, 'While one channel transfers angular momentum from the ferromagnetic layer (FM) to the TMD, the other channel returns angular momentum from the TMD to the YIG,' seems too simplistic. It will be better if you can elaborate on the discussion on this point. They should also discuss the effect of interfacial transparency and change in Meff (as Meff is quite different for different samples in Table 1) with A/P ratio on the damping parameter. However, as they mentioned in the discussion there is no existing model or theory that can explain their results. The photon flux experiment seems convincing, and the manuscript may lead to more work in this direction. Considering the major improvements in their manuscript and added theoretical support, I think the manuscript could be considered for publication.

Reviewer #2

(Remarks to the Author)

In this study, the authors investigate light-induced spin pumping in YIG/MoS₂ heterostructures and examine the roles of metallic edge states and semiconductor area states in spin current injection. The revised manuscript addresses most of the reviewers' comments. However, I would consider accepting the manuscript for publication only if the following key points are adequately addressed:

1. Fermi Level Discrepancy in Fig. 2(a): The authors present the local density of states (LDOS) for a MoS₂ triangular flake (57 nm lateral dimension, zigzag-terminated edges) in Fig. 2(a), with the Fermi level ($E=0$) positioned near the valence band maximum. This appears to contradict the typical behavior of the Fermi level within the band gap, as reported in previous studies (e.g., Nat. Commun. 6, 6293 (2015); J. Phys. Chem. C 118, 30309–30314 (2014)). The authors should clarify this discrepancy.

2. Impact of Violet Light on MoS₂ Properties: The authors suggest that violet light enhances the metallic state contribution in MoS₂. However, violet light can affect MoS₂ in several ways:

- (i) It may induce bonding between oxygen and Mo (or S) atoms, leading to the formation of an oxidation layer (ACS Appl. Mater. Interfaces 10, 27840–27849 (2018); Nano Res. 13, 358–365 (2020)).

(ii) It may also alter surface bonding or defect states, resulting in changes to the band structure (J. Nanoscience Nanotechnol. 20, 6500–6504 (2020)).

These light-induced effects should be discussed in greater detail to ensure a comprehensive understanding of the impact of violet light on MoS₂.

3. Minor Point: In lines 195–196 on page 7, the cited references should be properly listed in the References section.

Reviewer #3

(Remarks to the Author)

The revised manuscript titled “Light-driven disentanglement between metallic edge and semiconductor area states in MoS₂ monolayer flakes,” authored by Rodrigo Torrão Victor et al., presents significant findings on spin-pumping dynamics. The main results are novel, and have barely been presented in the literatures. This study investigates YIG/MoS₂ heterostructures and their function in spin-to-charge interconversion, which is crucial for spintronic device applications. The research highlights that MoS₂ monolayer flakes inject spin current through two primary mechanisms: metallic edge states and semiconductor area states. These mechanisms interact differently based on flake size, influencing spin-pumping behavior. Using density functional theory (DFT), the study calculates the local density of states, showing that light intensity and wavelength can regulate spin current injection. These findings suggest that light-driven spin current injection can be enhanced, diminished, or toggled, highlighting the potential for energy-efficient opto-spintronic devices. However, major revisions are still needed to enhance clarity.

1. The authors should explicitly state that merged edges or grain boundaries were not considered when estimating the perimeter. Additionally, recent literature, such as [H. Ahn, et al. Nat. Nanotech. 2024, 19, 955-961], shows that mirror twin boundaries in 1H-phase TMDs can serve as metallic channels. The manuscript should include explanations to rule out these effects.
2. References should be cited for statements derived from the literature; for instance, Line 64’s claim regarding flake morphology under sulfur-rich growth conditions should be supported by sources like [T. H. Yang, et al. Nat. Electron. 2024, 7, 29-38]. This should also apply to Line 167.
3. On Lines 155 and 160, the full forms of NM and IREE should be provided when first mentioned. Authors must ensure consistent abbreviation usage throughout.
4. The right branch of the V-shaped curve in Fig. 1e lacks sufficient data points, making it difficult to confirm if the trend is linear. The authors should add more data points to strengthen the analysis.
5. As indicated by eq.1, α_{sp} is not dependent on the NM layer’s perimeter. If possible, the authors should modify the equation to reflect this and provide a complete view of the two-phase competition.
6. Fig. S4’s “A₁” peak label should be corrected to “A_{1g}” for consistency with standard notation.
7. The paper’s structure needs improvement. Figures must be cited in sequence to facilitate smoother reading. For example, Line 162 mentions Fig. S15 without preceding Figs. S1-S14, disrupting the logical flow. Similarly, Line 354 discusses Fig. 4c before 4a and 4b. Each figure should be introduced in order, and all must be explained when first cited. The supplementary information should align with the main text.
8. Reference citation methods should be consistent (see Lines 195-196). Combining Figs. 2 and 3 into a single figure may improve coherence. Additionally, “ME” should be defined in Fig. 4. In Fig. 4b, an explanation is needed for why light-driven enhancement in Sample 2 is smaller than in Sample 1—possibly due to fewer sites for metallic excitation linked to a smaller flake area.
9. The sentence on Lines 393-395 describing colored arrows should be repositioned earlier, ideally before the discussion of sub-figures 4c-f. Line 420’s reference to “MoS₂” should be checked, as it might be a typo and should potentially read “NbSe₂.”
10. While DFT calculations clarify how light influences spin pumping, visualizing spin injection pathways through activated metallic sites would enhance comprehension. Authors are encouraged to depict these pathways for different cases using DFT simulations.

Reviewer #4

(Remarks to the Author)

Version 3:

Reviewer comments:

Reviewer #1

(Remarks to the Author)

I appreciate the significant efforts made by the authors to address the comments provided by reviewers. The manuscript has improved significantly in clarity and experimental details. Adding supporting measurements and more detailed explanations provide a stronger foundation for the work. Considering the improvements made in the manuscript, I think that the manuscript is now suitable for publication.

Reviewer #2

(Remarks to the Author)

The authors revised the paper based on the comments by the referees. The revised manuscript is much understandable and persuasive. I recommend the publication.

Reviewer #3

(Remarks to the Author)

The authors have addressed most concerns raised by the referees. The underlying mechanism for the competition between the metallic and semiconducting states remains ambiguous, the phenomena based on YIG/MoS₂ presented by the authors appear to be robust and repeatable, which probably can apply to other TMD materials. Considering that the substantial impact that this work may bring about to the spintronics field, I can therefore recommend its publication in Nature Communications. Before that, please ensure to correct the following minor typos and add explanations.

1. Following my previous comments on the notation of "A_{1g}", the authors should also correct E' to E_{2g}¹ mentioned in the same caption. Other than the notation issue, the measured two peaks red-shifted by ~6 cm⁻¹ toward lower wavenumbers compared with the literatures. The referee 's inference is that the authors didn't do calibration for the MoS₂'s peaks with reference peaks, for instance, peaks from the used substrate. Please explain.

2. In line 239 in Supporting Information, Table S3 seems a typo and should be Table 1.

Reviewer #4

(Remarks to the Author)

Subject: Reconsideration of Manuscript Titled "Edge and 2D States Disentanglement and Light-Driven Spin Injection in MoS₂"

Dear Reviewers,

We would like to express our gratitude for the time and effort you have invested in providing feedback on our manuscript titled "Edge and 2D states disentanglement and light-driven spin injection in MoS₂." We understand and respect your decision.

We are writing to request a reconsideration of the referee's decision, as we are fully prepared to address the issues raised by conducting the necessary work. We have revised the manuscript and included ab-initio calculations based on Density Functional Theory. This new model comprehensively explains the role of light in enhancing the metallic phase (edge states) at the expense of the semiconductor phase (area states). We are confident that this theoretical framework will address the most pressing issues raised by the reviewers and enhance the novelty of the manuscript.

While we acknowledge the concerns raised by the referees, particularly regarding the lack of model/theory and the clarity of the explanation, we firmly believe that our work presents novel findings with substantial implications for the field. The light-driven disentanglement of metallic edge and semiconducting area states in spin-pumping, as well as the light-tunable control of spin injection, are unprecedented and open avenues for both fundamental understanding and practical applications.

In response to the referees' comments, we have revisited the points raised in the reviews one by one, with special attention to those regarding the lack of a model/theory. The manuscript has been modified to address the main criticisms and to emphasize the main achievements of the work. Additionally, we have modified the manuscript title to better represent the changes in this new version.

We are confident that this revised version can resolve the technical concerns and provide the clarity requested by the reviewers.

Yours sincerely,

Flávio Garcia

Reviewer #1

The manuscript by Rodrigo et al. investigates the disentanglement of edge and 2D states and light driven spin injection in MoS₂. They observed two contributions to spin pumping: (a) from metallic edge states and (b) from 2D semiconductor states. By varying the thickness of MoS₂, they determined that spin current injection by the 2D semiconductor states is purely an interfacial effect, following the inverse Rashba-Edelstein effect. This finding, however, is not novel and has been reported in several previous works. Additionally, they demonstrate that light-driven spin current injection can manipulate spin pumping depending on the size of the MoS₂ flakes, which affects the edge states. While this manuscript is interesting in its current form, it does not provide a substantial advance required for Nature Communications. The authors may consider the following comments before submitting elsewhere:

(i) “This finding, however, is not novel and has been reported in several previous works”

We must agree with the referee that previous works exist about the spin injection into almost insulator MoS₂ layer. However, there is still controversy surrounding this topic. In the context of our study, it is crucial to determine the microscopy origin of the spin pumping. Initially, we chose to present this discussion in the manuscript for these reasons. They do not simply confirm existing results from the literature for similar systems but highlight a point raised, which is far from our main findings, namely, the competition between the metallic edge and the semiconductor area channels.

Additionally, to our knowledge, it's the first time that the spin pumping from an edge state has been reported. Moreover, the light-driven metallic states, which allow us to finally finely control the spin current injection and the spintronic behaviors, which is completely new, open several opportunities for low energy-consuming devices, etc. Therefore, we have made changes in the manuscript, condensing the main discussions and moving the detailed discussion of the REE in the 2D semiconductor states -which is not the novelty of the manuscript- to the supplemental material.

(ii) *While this manuscript is interesting in its current form, it does not provide a substantial advance required for Nature Communications.*

Indeed, the section addressing spin injection from YIG into the semiconductor portion of MoS₂ does not present a groundbreaking advancement compared to the current literature. However, the specific case of triangular monolayer flakes had not been previously reported. However, we understand that this aspect alone may not meet the prestigious standards of *Nature Communications*.

Beyond the novel concept of disentangling edge-state and area contributions in monolayer flakes, alongside the detailed fabrication and comprehensive characterization, the key experimental findings we report are both significant and novel:

We have identified two distinct channels contributing to spin pumping, each with a different microscopic origin. The semiconductor phase is driven by the inverse Rashba-Edelstein effect (iREE), while the metallic phase likely arises from the spin Hall effect (SHE). This dual contribution opens up two promising directions for these systems—either in device applications or as fundamental prototypes for deeper studies in spintronics.

Our experimental evidence clearly demonstrates that these two channels operate concurrently. While one channel transfers angular momentum from the ferromagnetic layer (FM) to the TMD, the other channel returns angular momentum from the TMD to the YIG. This competition gives rise to a compensation point where these spin-pumping contributions cancel each other out. It is crucial to note that although we can offer some plausible insights into the underlying mechanisms, no existing model or theory, to the best of our knowledge, adequately explains this phenomenon. We believe that a study presenting such clear, reproducible, and highly unexpected results—results that challenge conventional thinking and stimulate new questions for the scientific community—is indeed aligned with the caliber of *Nature Communications*.

Additionally, the ability to control the spintronic properties of this system—amplifying, attenuating, switching on, or completely turning off spin pumping—simply by modulating the intensity of incident light, represents a breakthrough innovation. This capability not only introduces a previously unreported mechanism in the field but also opens the door to a wide array of energy-efficient optoelectronic devices.

We believe these results alone justify the publication of our work in *Nature Communications*. Last but not least, in collaboration with a theoretical group, we have further strengthened the manuscript by developing a model based on density functional theory (DFT) calculations. These calculations provide a solid theoretical foundation for understanding the balance between the semiconductor and metallic phases through illumination with light of tailored wavelength.

In light of the original results, the additional modifications, and the new DFT-based insights, we are confident that our work meets the high standards required for publication in such a prestigious journal.

1. The introduction part could be improved by providing a more detailed discussion on metallic edge states, as this forms the main result of the work. The other parts of the introduction could be made more concise.

We thank the referee for the comments and the time dedicated to reviewing the manuscript. We have improved the introduction by emphasizing the new calculations included in the manuscript. Our approach involves controlling the balance between the metallic and semiconductor phases in the flakes and, consequently, the balance between the two channels for spin pumping. We achieve this control by illuminating the system and adjusting the light intensity with the appropriate energy (wavelength).

2. Have the authors tried preparing samples with metallic ferromagnets instead of YIG and measuring spin-to-charge conversion? It would be interesting to see if spin-to-charge conversion follows the V-shaped dependence similar to Fig. 1e.

YIG, as an insulating ferromagnet, has several advantages relative to metallic ferromagnets. It eliminates the influence of the anisotropic magnetoresistance on the FMR signal and the spin pumping. Otherwise, as occurs with usual metallic ferromagnets such as FeNi, CoFe, etc., in FM/NM bilayers, one has to make measurements rotating the sample to extract by fitting the right spin pumping contribution. This additional data treatment can introduce artifacts to the spin pumping measured. Besides this, YIG exhibits the smaller α value among the ferromagnets allowing a more precise α variation with and without the NM layer.

3. The enhancement in damping due to spin pumping (α_{SP}) shows a finite value for all points in Fig. 1a (edge length dependence) and 1b (area dependence). How can α_{SP} be zero as shown in Fig. 1e? Please cite previous works, if available, showing this effect.

The enhancement in damping is due to the presence of MoS₂ flakes deposited over the YIG. An α_{SP} value as zero means that α_{YIG} , the damping of the pure YIG, does not change with the MoS₂ coverage. It is shown in equation 1 of the manuscript, i.e., $\alpha_{SP} = 0$ means $\alpha_{YIG} = \alpha_{YIG/MoS_2}$. The text was rewritten to clarify this point. However, as discussed in the manuscript, it is more complicated than relating α to the flake perimeter or area; it scales with area/perimeter, as seen from experimental data. As supported by the new results of DFT calculation and the experiments performed with the samples illuminated, α depends on both the area (semiconductor) and edge (metallic) state contributions, which have opposite contributions to α .

4. The origin of the compensation point requires more discussion. The authors may need to measure more samples near the compensation point to demonstrate that α_{SP} can indeed be zero.

The compensation point is an extrapolation. In general, MoS₂ coverage enhances α , but for a given area/perimeter, an equal contribution of the flake area (semiconductor) and edge (metallic) states can occur, providing α SP zero. However, fabricating new samples with a suitable $A_{\text{Total}}/P_{\text{Total}}$ ratio to precisely match the compensation point on the V-curve is a challenging task. However, we present an alternative approach to investigate the V-shaped curve, mainly the compensation point region. By using light and varying its intensity and wavelength, one changes the ratio between the area and edge states passing by the compensation point. In such a way, we can easily play with both contributing channels to the spin-pumping, namely, the semiconducting and metallic ones, enabling us to access the compensation point precisely.

5. Including simulations or theoretical modeling would help in understanding the results.

In this new version of the manuscript, we have included DFT calculations. We have calculated the local density of states (LDOS) projected onto Mo and S atoms at both the edge and area positions. The edge states were arbitrarily defined as the flake's two outermost atomic lines, while all other sites are considered as the area states. Additionally, the partial density of states (PARCHG) can provide insight into the spatial localization of electronic states. This approach helped to understand the spin pumping mechanisms in TMD small flakes, also when driven by light, which was never observed before.

6. As a final comment, in light of the original results we are presenting, mainly the spin pumping driven by light, the additional DFT calculations showing the disentanglement of the semiconductor area, and the metallic edge contributions of the MoS₂ flakes, which provide a substantial advance to the spintronics field, we are confident that our work meets the high standards required for publication in Nature Communications.

Reviewer #2

In this study, the authors examined the efficiency of spin current injection in MoS₂ monolayer flakes on YIG substrates through spin pumping measurements. They explored how the Gilbert damping factor α changes with the $A_{\text{Total}}/P_{\text{Total}}$ ratio by varying the size of the MoS₂ monolayer flakes. Their findings suggest that both the 2D semiconductor properties and the metallic edge states of MoS₂ contribute to the spin injection process. They also show that light-driven spin pumping can enhance, reduce, or even completely suppress spin injection in the YIG/MoS₂ heterostructures.

The findings, while potentially valuable, do not present a breakthrough or novel insight that significantly advances the field of spintronics or 2D materials science. One key concern is the unclear reasoning behind the use of the $A_{\text{Total}}/P_{\text{Total}}$ ratio to explain the behavior of the Gilbert damping factor. The relationship between these parameters and the spin dynamics in the system is not sufficiently justified or explained, leaving readers with unanswered questions about the underlying mechanisms. Even after revisions to address the issues, the work may still not meet the high bar set by Nature Communications. The manuscript may be better suited for publication in other materials science journal after revision.

We would like to take this opportunity to emphasize the significance and novelty of our findings, particularly regarding the control of spintronic effects using light, besides separating the contributions of area and edge to the spin pumping. Our data demonstrate that by adjusting light intensity, we can modulate these effects precisely—either amplifying, diminishing, switching them on, or even completely turning them off. We believe this ability to control spintronic effects finely could represent a substantial advancement in the field of spintronics, and we are confident it can be considered a genuine breakthrough.

We understand that the referee has raised some concerns, particularly about the absence of theoretical models to explain the observed effects. As we all recognize, physics is a deeply empirical science where models are expected to explain experimental observations rather than the other way around. In our view, when experimental results challenge existing models and provoke unanswered questions—albeit temporarily—this is often a hallmark of a true scientific breakthrough. Nevertheless, anticipating these concerns, we have collaborated with a theoretical group to develop a comprehensive theory that explains the influence of light on enhancing the metallic phase (edge states) in our system at the expense of the semiconductor phase (area states). We believe that this theoretical framework included in this new version of the manuscript will address the most pressing and relevant questions raised by the referee.

Reviewer #3

Overall, this is an interesting work. The goal of this work was to disentangle effects of the metallic edge and the semiconducting states on the spin jumping behaviours. As promising as it is, I suggest a major revision and hope that the following concerns could be carefully addressed before I can decide whether or not recommend its publications.

1. Please provide the values that are used for each term in Eq. 1 to obtain Fig. 1a and b, and cite references where they were originally mentioned if necessary.

We thank the referee for the comments and the time dedicated to reviewing the manuscript. To address this, the paragraphs before and after Equation 1 have been rewritten. In addition, Section S3 in the Supplementary Information has been improved to clarify the origin of the values used in Figure 1 and to provide the important values obtained and used in all ferromagnetic resonance and spin pumping measurements.

2. The scale bar in Fig. S4 is missing. Please add it.

The Figure S4 has been slightly modified by adding the scale bar, which is also explained in the figure caption.

3. If the $g^{\uparrow\downarrow} \approx (SkF^2)/(4\pi)$ as written in p. 5, and there is another S term in the later bracket of Eq.1, the α SP seems not proportional to the S (i.e., the YIG/MoS₂ contact area). Please clarify why a linear correlation can be expected. Also, it seems that an equation (or even a theoretical model) to express the relation between α SP and the Perimeter term was not provided. Therefore, the referee felt confused while reading the first paragraph in p. 6. Please clarify this.

The first paragraph on p. 6 has been rewritten to avoid misinterpretation. Regarding the area dependence in Equation 1, the authors agree with the reviewer that the term S in brackets is canceled by the term S resulting from $g^{\uparrow\downarrow}$, also inside the brackets. However, there is another term, $g^{\uparrow\downarrow}$, outside the brackets, which, when expanded, makes the area (S) dependence of SP clear. As an additional effort to make this clear to readers, a discussion and derivation of this equation showing the S-dependence of SP has been added in Section S3 of the Supplementary Material, and the equation with a clear form of the S dependency was added to the equation 1.

4. How did the authors estimate the total perimeter of MoS₂ to obtain Fig.1a. As mentioned in Supplementary Information, the coverage of CVD-grown MoS₂ looks tunable. The referee wonders what kind of sample was used, i.e., flakes or films, for Fig. 1a. Did authors include the merged edge in the estimation? If the MoS₂ sample was a continuous film, how did the authors distinguish where the edges exactly are given that the contrast of merged boundaries should be extremely weak under OM.

The total perimeter (and area) of each sample was carefully evaluated by mapping and measuring the edge of all the flakes in the sample in direct contact with the YIG. Since the optical contrast of the flakes is very weak, the mapping was performed by optical microscopy using different objective lenses and filters to enhance the contrast of the flakes concerning the substrate. After mapping, the edge length of

the flakes was estimated using ImageJ software. A subsection (S1.9) was added to the supplementary material to clarify these points. We have also added subsection S1.7 to improve the understanding and visualization of the samples. We agree with the reviewer that the merged edges show weak contrast under light microscopy. However, the merged edges were not used to obtain the total edge length. This means that the region where two (or more) different flakes merge to form a larger flake is not included in the quantification of the total edge length. Therefore, although the optical contrast is very weak in the boundary edges, it is still possible to visualize the boundaries of the flakes. Specifically, for the film, the edge length was obtained from a very small region of the YIG not covered by the MoS₂ film.

5. Please explain what those unspecified data points are in Fig. 1e. it would be helpful if the authors specify and provide the appearance of each sample used.

The highlighted data in Figure 1e are the samples used in the study of the influence of light on spin pumping. To emphasize this, a sentence has been added to the caption of Figure 1. The authors agree with the referee, and a section (S1.8) has been added to the Supporting Information to provide the appearance of each sample used in this work.

6. In Fig. 2c-f, the figure captions for “ME”, ”SP”, “SC”, and “M” are missing.

A sentence has been added to the caption in Figure 2 to include the missing abbreviations and to improve the overall clarity of the figure.

7. In p. 6, the authors stated as follows, “When electrons are excited from the valence to conduction bands by violet light, the metallic contribution is amplified,...”. This might not be reasonable since there is no band gap for metallic materials. Through the entire Fig. 2, the influence of light shining on the MoS₂/YIG samples were not clear. The referee’s inference is that the light affects both the semiconducting and metal states, and the photon flux also promote the spin injection from the YIG to the MoS₂. However, if the light enhanced the metal state in S1 to S4, why the enhancement improved α_{SP} for S1&2 while it decreased the α_{SP} value for S3&4. Would that be possible for the intensified light to only enhance the semiconducting states for all the four samples? Please verify the mechanisms, supported with theoretical model in the main text.

To provide a better explanation of this subject, we have included DFT calculations. We have calculated the local density of states (LDOS) projected onto Mo and S atoms in both edge and area positions. The edge states were arbitrarily defined as the two outermost atomic lines of the flake, while all other sites are considered as the area states. In addition, the partial density of states (PARCHG) can provide insight into the spatial localization of electronic states. This approach helped to

understand the spin pumping mechanisms in TMD small flakes, as well as when they are driven by light. This theoretical framework and calculations have been included in the manuscript to support and clarify the discussion. One of the results of these calculations is the increase in the metallic states of MoS₂ when excited with energies consistent with those used in the experiments. The light-induced states can increase the SP in samples S1 and S2 while decreasing the SP in samples S3 and S4 because two channels are competing for the total SP, while samples S1 and S2 have the SP mainly due to the metallic states and therefore increased by the light, samples S3 and S4 have the SP mainly due to the semiconducting states and therefore decreased by the light.

8. Following Q8, it would be helpful if the explanation can be supported with band diagrams.

The band diagrams calculated by DFT were included in the manuscripts along with the local density of states (LDOS) projected onto Mo and S atoms located at the edge and area, and the partial density of states (PARCHG), which provided insight into the spatial localization of the electronic states.

9. It has been well known that monolayer MoS₂ has a direct band gap 1.8~1.9 eV. The authors mentioned a different value of 2 eV, which was even believed an underestimated value. The referee wonders based on what the authors think the monolayer MoS₂ used in this work had a larger band gap. In addition, literatures also have shown that a red-light laser (with a wavelength ~633 nm) is resonant to the band gap of monolayer MoS₂ and enough to excite electrons, (which is why we can see the A-exciton peak in photoluminescence). The origin that accounts for the observation regarding Fig. 2g might need to be reconsidered.

It is well known that the MoS₂ monolayer has a direct band gap in the K-point of the Brillouin zone. It is also well known that the optical bandgap is between 1.8 and 1.9 eV, therefore resonating in the absorption of a red-light laser with a wavelength of about 633 nm and thus giving the A exciton peak in the photoluminescence. However, it is important to note that there is a difference between the optical bandgap and the electronic bandgap. The electronic bandgap (E_g) is characterized by a single particle or quasiparticle excitation and is defined by the sum of the energies to tunnel an electron and a hole separately. In other words, the energy required to move an electron from the valence band to the conduction band. The optical band gap (E_{opt}), on the other hand, describes the energy required to create an exciton (a correlated two-particle electron-hole pair) by optical absorption. The difference between these energies ($E_g - E_{opt}$) gives the exciton binding energy (E_b).
[DOI: 10.1038/NMAT4061]

In conventional semiconductor materials such as Si, Ge, GaAs, etc., the exciton binding energies are in the order of a few meV and are sometimes neglected. [DOI: 10.1103/physrevb.29.1807 ; DOI: 10.1103/PhysRevB.24.1134]

On the other hand, the 1H TMDs, such as the MoS2 monolayer, present several properties (weak dielectric screening, strong geometrical confinement, strong Coulomb interaction, etc.), which together lead to several different types of exciton formation and binding energies in the order of 0.5 eV, which cannot be neglected [DOI: 10.1038/s41699-018-0074-2]. Therefore, photoluminescence is used to determine the optical bandgap rather than the electronic bandgap and is not an accurate representation. In fact, until recently, there was a lack of data to directly probe the electronic bandgap of TMDs, which is no longer the case, as can be seen in the following references. [DOI:10.1038/ncomms7298; DOI:10.1103/PhysRevLett.113.026803; DOI: 10.1103/PhysRevLett.113.076802 ; DOI: 10.1038/nmat4061 ; DOI: 10.1038/s41699-018-0074-2]

Part of this confusion arises from the coincidence of the experimental values of the optical bandgap through photoluminescence and the calculated values obtained by Density Functional Theory (DFT). However, it is well known that conventional DFT strongly underestimates the band gaps, usually due to the local density or gradient approximations - for example, the gap estimated by DFT for bulk MoS2 is typically 0.76 eV, which is significantly smaller than the experimental values, which are of the order of 1.3 eV. To obtain a more accurate estimate of the value of the band gap, a quasi-particle approach in the GW approximation is often used - with this approach the gap for bulk MoS2 is found to be 1.3 eV, in agreement with the experimental value.

For the case of the MoS2 monolayer, the values for the electronic gap using the GW method were found to be in the range of 2.76 eV, which is ~0.9 eV higher than the typical photoluminescence experimental results. However, this difference is explained by the high binding energy of the inherent 2D screening of the Coulomb interaction in the monolayer. Therefore, better estimates of the electronic band gap of the MoS2 monolayers should be provided by the GW approximations, which yield a value of 2.76 eV, and/or electronic measurements, which found an electronic band gap of 2.4 eV for the monolayer, and also considering that the substrate can also influence in the exact values of the band gap. [DOI: 10.3390/cryst6110143; DOI: 10.1103/PhysRev.139.A796; DOI: 10.1038/srep29184; DOI: 10.1002/andp.201400128]

In this sense, we have added section 2 in the supplementary material with a similar explanation. In addition, in subsection S1.7, we present the measurement of light absorption as a function of wavelength for our samples. It is possible to see a very intense absorption peak at ~415 nm, which is in the same region as the violet light

used in the discussions in the main text. Furthermore, the absorption intensity in the violet region is at least two times higher than that in the red-light region (with a wavelength of ~633 nm). These experimental results and the above considerations support the possibility that the violet light promotes electrons from the valence band to the conduction band in the MoS₂ monolayers.

10. In p. 11, the full name of SOT should be provided.

On page 11, the abbreviation has been changed to the full name (spin-orbit torque).

Dear Reviewers,

We would like to extend our sincere gratitude for the time and effort you have dedicated to reviewing our manuscript titled "Light-driven disentanglement between metallic edge and semiconductor area states in MoS₂ monolayer flakes." We deeply appreciate your thoughtful and thorough evaluation of our work. The insightful comments and constructive suggestions provided have been invaluable in strengthening the quality of the manuscript.

We are pleased to submit the revised version of the manuscript, along with a detailed point-by-point response to each reviewer's comments. In this updated version, we have made several revisions to address the feedback and improve the clarity and rigor of the manuscript. For your convenience, the modifications in the main manuscript and in the supplementary information are highlighted in red.

We believe the revisions effectively address the technical concerns and provide the additional clarity requested. We look forward to your feedback and hope the changes meet your expectations.

Once again, thank you for your thoughtful review of our manuscript.

Yours sincerely,
Flávio Garcia

Reviewer #1 (Remarks to the Author):

The revised manuscript by Rodrigo et al. have tried their best to address the comments by all the reviewers. They have added supporting measurements and experimental details. They have developed theoretical background which tries to explain the influence of light on enhancing the metallic phase in their system at the expense of the semiconductor phase. They have modified the manuscript to highlight the novelty in a better way, which is the modulation of the balance between metallic and semiconductor channels using lights of suitable wavelength. This could help control the spin current injection in spintronic devices. I believe the manuscript has improved quite a lot, and many of the doubts have been cleared.

However, there are still a few unanswered questions, especially regarding the mechanism. Their statement, 'While one channel transfers angular momentum from the ferromagnetic layer (FM) to the TMD, the other channel returns angular momentum from the TMD to the YIG,' seems too simplistic. It will be better if you can elaborate on the discussion on this point. They should also discuss the effect of interfacial transparency and change in M_{eff} (as M_{eff} is quite different for different samples in Table 1) with A/P ratio on the damping parameter. However, as they mentioned in the discussion there is no existing model or theory that can explain their results. The photon flux experiment seems convincing, and the manuscript may lead to more work in this direction. Considering the major improvements in their manuscript and added theoretical support, I think the manuscript could be considered for publication.

R: We would like to thank Reviewer #1 for their time and effort spent during the review process, as well as for the thoughtful suggestions and positive feedback. We greatly appreciate the reviewer's constructive comments, and we take this opportunity to address each point in more detail.

1- *Their statement, 'While one channel transfers angular momentum from the ferromagnetic layer (FM) to the TMD, the other channel returns angular momentum from the TMD to the YIG,' seems too simplistic. It will be better if you can elaborate on the discussion on this point.*

R: The authors thank the reviewer for the opportunity to improve the discussions about the mechanisms that lead to spin-to-charge interconversion. To address this point, we modify the second paragraph of the discussion section, lines 510-529. First, we made minor modifications to this part. In the sequence, we added a discussion on how the competition between the metallic edge and semiconductor area state channels can occur, drawing parallels with well-known cases of competition for spin current injection to heavy metallic materials with high spin-orbit coupling such as Pt and W.

2- *They should also discuss the effect of interfacial transparency and change in M_{eff} (as M_{eff} is quite different for different samples in Table 1) with A/P ratio on the damping parameter.*

R: The authors thank the reviewer for the suggestion to include a discussion on interfacial transparency. In lines 263-269, we have introduced a discussion on spin mixing conductance

($G_{\uparrow\downarrow}$) and interfacial transparency. While the influence of the magnetic parameters of YIG on the analysis of $G_{\uparrow\downarrow}$ is significant, it is not the primary focus of the manuscript. Therefore, we refer readers to Supplementary Information Section 3 for a more detailed discussion on this topic.

Regarding the M_{eff} , the 13 samples used in this study exhibit values ranging from 1839 to 2310 G. Since the equation for $G_{\uparrow\downarrow}$ (Eq. S7) depends on several constants, including M_{eff} , thickness, and γ_{eff} , and since the thickness is uniform across all samples and γ_{eff} is nearly identical for all, the variations in M_{eff} lead to small fluctuations in the V-shaped curve. The dependency of $G_{\uparrow\downarrow}$ on the area-to-perimeter ratio is presented in Supplementary Information Section 4, Figure S16. When comparing the adjustment of the modulus of a straight line for both α_{SP} (Figure 1(e)) and $G_{\uparrow\downarrow}$ (Figure S16), the dispersion for the case of mixed spin conductance with a value of $R^2 = 0.9185$ is even smaller when compared to $R^2 = 0.9002$ in the case of alpha SP. A more comprehensive discussion can also be found in Supplementary Information Section 4, lines 236-246.

Reviewer #2 (Remarks to the Author):

In this study, the authors investigate light-induced spin pumping in YIG/MoS₂ heterostructures and examine the roles of metallic edge states and semiconductor area states in spin current injection. The revised manuscript addresses most of the reviewers' comments. However, I would consider accepting the manuscript for publication only if the following key points are adequately addressed:

R: The authors thank Reviewer #2 for his careful reading and positive feedback.

1. Fermi Level Discrepancy in Fig. 2(a): The authors present the local density of states (LDOS) for a MoS₂ triangular flake (57 nm lateral dimension, zigzag-terminated edges) in Fig. 2(a), with the Fermi level ($E=0$) positioned near the valence band maximum. This appears to contradict the typical behavior of the Fermi level within the band gap, as reported in previous studies (e.g., Nat. Commun. 6, 6293 (2015); J. Phys. Chem. C 118, 30309–30314 (2014)). The authors should clarify this discrepancy.

R: The apparent discrepancy regarding the Fermi level position compared to the cited articles is due to different electronic structures. The cited articles investigate different situations, such as defects in MoS₂ monolayers, Sulphur vacancies, and adatoms. In our case, the density of states calculated for triangular flakes has similar behavior to DFT calculations for zigzag terminated MoS₂ nanoribbons; PNAS 113, 8583–8588 (2016). These nanoribbons show metallic edge states, and their Fermi level is located near the valence band maximum as well. Thus, this apparent discrepancy in the Fermi level is due to different conditions.

2. Impact of Violet Light on MoS₂ Properties: The authors suggest that violet light enhances the metallic state contribution in MoS₂. However, violet light can affect MoS₂ in several ways:

(i) It may induce bonding between oxygen and Mo (or S) atoms, leading to the formation of an oxidation layer (ACS Appl. Mater. Interfaces 10, 27840–27849 (2018); Nano Res. 13, 358–365 (2020)).

(ii) It may also alter surface bonding or defect states, resulting in changes to the band structure (J. Nanoscience Nanotechnol. 20, 6500–6504 (2020)).

These light-induced effects should be discussed in greater detail to ensure a comprehensive understanding of the impact of violet light on MoS₂.

R: The effect of illumination on the samples has been a concern since the beginning of the study. For example, it is known that the MoS₂ flakes can be destroyed during electron microscopy, depending on the current and voltage of the beam. In order to ensure that violet illumination does not alter the properties of YIG/MoS₂, especially the magnetic properties, we have adopted a protocol for all samples to measure the ferromagnetic resonance before any violet illumination of the sample and compare it with the ferromagnetic resonance of the samples after the exposures, measured without illumination. It is important to note that the ferromagnetic resonance of the YIG/MoS₂ system is very sensitive to any defects and structural and electronic changes. Therefore, if any irreversible changes occur in the YIG/MoS₂ samples, their ferromagnetic resonance should also show some changes. We have already commented on this in lines 460-465, where we mention the Supplementary Information section S4, which presents the ferromagnetic resonance measurements before and after light exposure and discusses these results. It is also important to note that the increase or decrease in spin pumping during the violet light exposure is fully reversible, and it is possible to observe these variations several times. In addition, the light-induced spin pumping measurements were not performed simply by increasing or decreasing the light intensity. The data presented in Figure 3 (note that the figure number has been changed as other reviewers suggest merging the previous Figures 2 and 3) were obtained in a non-sequential order of light intensity. This protocol of obtaining data in a non-sequential order further reinforces the absence of light damage in the samples. The aforementioned effects of ultra-violet (UV) light in MoS₂, namely the bonding between oxygen and molybdenum/sulfur atoms, the formation of an oxidation layer, surface bonding, and defect states leading to changes in the band structure of MoS₂, are all irreversible. They were, therefore, incompatible with the reversible behavior observed in the measurements of our samples.

Nevertheless, the authors agree with the reviewer that a more detailed discussion in this regard could improve the overall quality of the manuscript, and appreciate the reviewer's suggestions and opportunities for improvement. In the main manuscript, lines 461-468, we have introduced a discussion about the effects of UV light incidence in MoS₂ observed in the mentioned references while pointing out the experimental differences between our case and the literature. In Supplementary Information section S5 (previously section S4), we have increased the explanation of the protocol for measuring the ferromagnetic resonance before and after the light exposure in lines 202-207. A comment about the non-sequential data acquisition protocol for the light-induced spin pumping measurements was also added in lines 296-301.

3. *Minor Point: In lines 195–196 on page 7, the cited references should be properly listed in the References section.*

R: The references in lines 195-196 on page 7 were properly formatted. They are mentioned in line 200 and shown in the reference section, pages 20-21, lines 655-670.

Reviewer #3 (Remarks to the Author):

The revised manuscript titled "Light-driven disentanglement between metallic edge and semiconductor area states in MoS₂ monolayer flakes," authored by Rodrigo Torráo Victor et al., presents significant findings on spin-pumping dynamics. The main results are novel, and have barely been presented in the literatures. This study investigates YIG/MoS₂ heterostructures and their function in spin-to-charge interconversion, which is crucial for spintronic device applications. The research highlights that MoS₂ monolayer flakes inject spin current through two primary mechanisms: metallic edge states and semiconductor area states. These mechanisms interact differently based on flake size, influencing spin-pumping behavior. Using density functional theory (DFT), the study calculates the local density of states, showing that light intensity and wavelength can regulate spin current injection. These findings suggest that light-driven spin current injection can be enhanced, diminished, or toggled, highlighting the potential for energy-efficient opto-spintronic devices. However, major revisions are still needed to enhance clarity.

R: The authors would like to thank the reviewer for the time spent reviewing our manuscript and for the valuable suggestions. We will address each comment in detail below, and to provide more detailed and accurate answers, some comments were splitted.

1. *The authors should explicitly state that merged edges or grain boundaries were not considered when estimating the perimeter. Additionally, recent literature, such as [H. Ahn, et al. Nat. Nanotech. 2024, 19, 955-961], shows that mirror twin boundaries in 1H-phase TMDs can serve as metallic channels. The manuscript should include explanations to rule out these effects.*

R: In response to the first suggestion, we have included the statement: "It is essential to emphasize that the MoS₂ merged edges and grain boundaries were excluded from the total perimeter estimate" in lines 184-185.

Additionally, we appreciate the reviewer for highlighting this recent literature, published online on July 3 of this year, during the manuscript's review process. The study indeed demonstrates that the metallic edge states in MoS₂ contribute to the interconversion of charge and spin currents, which opens up various avenues for exploring other electronic states, such as different edge states and the previously mentioned boundary grain state. However, we must highlight that we have not observed any experimental evidence regarding the impact of boundary grain states in our research. Furthermore, the referenced work identifies these intriguing 1D metallic

channels only under a specific configuration (60° rotation between crystals). In our study, the relative crystal orientation of the larger flakes was uncontrolled, making the establishment of such a precise scenario highly improbable. A diverse range of angles between crystals is anticipated, including, on rare occasions, this particular angle. As a result, we do not anticipate that these few instances would significantly influence our measurements. To address these points, we have added a discussion in lines 530-539 on the potential for other channels of spin-charge interconversion in TMDs and focusing on the specific case of 1H-MoS₂.

"It is noteworthy that the zigzag metallic edge states studied here may not be the only additional channel capable of spin-to-charge interconversion in TMDs. Recently, the existence of one-dimensional metallic states in 1H-MoS₂ grain boundaries has been shown. However, this particular configuration, called mirror twin boundaries, only occurs when the two adjacent crystals are exactly 60° rotated. In our case, the relative crystal orientation of the adjoining flakes was not controlled, so the formation of this specific case is unlikely. In addition, no experimental evidence for boundary influence on spin pumping has been found. Nevertheless, the existence of different one-dimensional metallic channels in TMDs opens the possibility of investigating new channels of spin-to-charge interconversion."

2. References should be cited for statements derived from the literature; for instance, Line 64's claim regarding flake morphology under sulfur-rich growth conditions should be supported by sources like [T. H. Yang, et al. Nat. Electron. 2024, 7, 29-38]. This should also apply to Line 167.

R: The statement in lines 64-65 regarding the MoS₂ morphology obtained under sulfur-rich growth conditions was supported by the recommended reference and others, including H. Schweiger, et al., Journal of Catalysis 2002, 1, 76-87 and H. Xu, et al., Nature Communications, 2016, 7, 12904. This was also done for the statement in lines 167-169.

3. On Lines 155 and 160, the full forms of NM and IREE should be provided when first mentioned. Authors must ensure consistent abbreviation usage throughout.

R: The 'NM' abbreviation on line 155 has been removed, and the complete form was used instead. The abbreviation 'IREE' stands for the inverse Rashba-Edelstein effect, which is the Kelvin-Onsager reciprocal of the Rashba-Edelstein effect (REE). The REE was previously defined in the Introduction section, line 43. We have added a sentence in lines 162-163, highlighted in red, to explain the relationship between these two effects and improve the manuscript's clarity. We also made minor changes in the manuscript (lines 272, 275, 277, 481, 482, 496, 499, and 513) to avoid duplicate definitions of abbreviations.

4. The right branch of the V-shaped curve in Fig. 1e lacks sufficient data points, making it difficult to confirm if the trend is linear. The authors should add more data points to strengthen the analysis.

R: We agree with the reviewer that more points in Figure 1e, especially in the right branch, would be helpful to strengthen the analysis. However, making new samples with an appropriate $A_{\text{Total}}/P_{\text{Total}}$ ratio to improve the number of samples in the right branch is a very challenging task. One of the reasons for using light and varying its intensity and wavelength was to overcome the experimental difficulty of making large area samples and transferring these large flakes into the YIG without damaging or darkening them. To clarify this, we have improved the commentary on this in lines 274-275. We would also like to emphasize that the data fitting shown in Fig. 1e was done with a single equation of the absolute value of a straight line utilizing all 13 data points (13 samples). In addition, if we fit only the left branch of the data, the linear trend persists, and, as expected, the parameters a and b (slope and constant) remain consistent under their respective uncertainties.

5. As indicated by eq.1, α_{sp} is not dependent on the NM layer's perimeter. If possible, the authors should modify the equation to reflect this and provide a complete view of the two-phase competition.

R: We agree with the reviewer that Equation 1, as proposed by Tserkovnyak, Brataas, and Bauer and which makes the basis of the spin pumping theory, does not fully reflect the behavior observed in our work. The behavior we observe should be described by an equation similar to the one we use to fit the data in Figure 1 (e): $\alpha_{sp} = \left| a \frac{A_{\text{Total}}}{P_{\text{Total}}} + b \right|$. However, a more theoretical approach is required to describe the physical properties included in the coefficients a and b . We are working on a model to calculate the spin pumping, including DFT, and taking into account the surface and edge states to address the two-phase competition. However, it remains undetermined whether we will achieve success. Consequently, we prefer to keep Eq. 1 unchanged and hope to address this point in an upcoming publication.

6. Fig. S4's "A_1^" peak label should be corrected to "A_1g" for consistency with standard notation.

R: The Raman peak label notation was changed to the standard notation "A_1g" in Figure caption S4 and the discussions in sections S1.3 and S1.4 (lines 49, 55, and 63) of the Supporting information.

7. The paper's structure needs improvement. Figures must be cited in sequence to facilitate smoother reading. For example, Line 162 mentions Fig. S15 without preceding Figs. S1-S14, disrupting the logical flow. Similarly, Line 354 discusses Fig. 4c before 4a and 4b. Each figure should be introduced in order, and all must be explained when first cited. The supplementary information should align with the main text.

R: To improve the reading quality and align the order of the Supplementary Information content with the order of the main manuscript, we have made some changes in both the manuscript and the Supplementary Information. In general, we no longer mention any figure in the Supplementary Information. Instead, we mentioned its sections. First, in line 133, we have introduced a direct mention of Section S1, which refers to the MoS₂ flakes' growth, transfer, and characterizations to evidence their properties and support their quality. In the sequence, we

have moved the Section that discusses the Rashba-Edelstein effect (the referred Figure S15 is included in this section) to Section S2, and we directly mention it in line 165. In lines 267-268, Section S3 is mentioned to address the details of the ferromagnetic and spin pumping. Further, in line 293, we directly mention Section S4, which discusses the MoS₂ monolayer band gap. This is followed by Section S5, which provides evidence that the light doesn't affect the magnetic properties of the YIG/MoS₂ system by measuring the ferromagnetic resonance of the system before and after exposing the samples to light. The section S5 is mentioned in line 470.

To further improve the reading quality, in lines 188-189, we have introduced Figures 1 (a) and (b); in line 354, we have introduced a brief description of Figure 3(a). In the sequence, lines 363 and 364, we have introduced Figure 3(b). To align with both the smoother reading suggested here and the following suggestion 9, we also presented in lines 363 to 371 a generic explanation regarding the diagrams in Figures 3(c) to 3(f), allowing us to avoid some repeated definitions of the diagram's elements in the further paragraphs.

8. (a) *Reference citation methods should be consistent (see Lines 195-196).*

R: The references in lines 195-196 on page 7 were adequately formatted to be consistent with the other references along the text. Also, they are now shown in the reference section, pages 20-21, lines 655-670.

8. (b) *Combining Figs. 2 and 3 into a single figure may improve coherence.*

R: Figures 2 and 3 have been combined into a single figure. In addition to this combination, we have made minor changes to improve readability and quality. First, we have combined the previous captions for Figures 2 and 3, adjusting some parts to avoid duplicate explanations or definitions. Second, in the new figure, we have used lowercase Roman numerals for panels A and B and uppercase Roman numerals for panel C, which improves the visualization of the flake mentioned in the figure. Thirdly, we have made minor adjustments to the manuscript text lines 275 to 351, mainly concerning the use of panel C instead of Fig. 3, and the paragraphs have been organized so that each panel is explained in a different paragraph to improve overall clarity.

8- (c) *Additionally, "ME" should be defined in Fig. 4.*

R: The abbreviation "ME" was included in the caption for Figure 4 (now Figure 3).

8- (d) *In Fig. 4b, an explanation is needed for why light-driven enhancement in Sample 2 is smaller than in Sample 1—possibly due to fewer sites for metallic excitation linked to a smaller flake area.*

R: The authors agree with the reviewer that an explanation of why the light-driven enhancement in the spin pumping of sample S2 is smaller than the one in sample S1 would improve the quality of the manuscript. In this regard, we have included a discussion in the lines 395-400. As discussed in Fig. 2 and mainly in Fig. 3 (now just Fig. 2), the light will excite metallic states in

the MoS₂ flakes, and depending on the size of these flakes, the probabilities of these occupations could be different, leading to differences in amplitudes in the light-driven spin pumping.

9. (a) *The sentence on Lines 393-395 describing colored arrows should be repositioned earlier, ideally before the discussion of sub-figures 4c-f.*

R: As mentioned in the previous answer, we have moved and improved the sentence in lines 395-400 to lines 363-371. Additionally, at this point, we discussed in detail the meaning of each arrow in the diagram. Specifically, red arrows indicate the metallic edge, and blue arrows represent the 2D semiconductor contributions in the opposite direction. The gray arrows represent the total spin pumping measured, which is the difference between the metallic edges and 2D semiconductor contributions. The purple arrows represent the light-driven contributions.

9- (b) *Line 420's reference to "MoS2" should be checked, as it might be a typo and should potentially read "NbSe2."*

R: To prevent any misunderstandings, we have changed the sentence to lines 434-435.

10. *While DFT calculations clarify how light influences spin pumping, visualizing spin injection pathways through activated metallic sites would enhance comprehension. Authors are encouraged to depict these pathways for different cases using DFT simulations.*

R: Our DFT calculations obtained the density of states for a MoS₂ triangular flake locally, i.e., site by site, and the integrated value over the whole flake. It allowed us to understand how the energy levels are occupied as a function of light excitation, affecting spin pumping. However, the spin pumping calculation is a much more complex task, which needs to consider many body interactions. We intend to do so in a forthcoming publication using linear response theory, combining DFT and effective Hamiltonians, which have been successfully used for the magnetization dynamics of 2D materials. Phys. Rev. B 102, 014450 (2020). A comment was added in lines 551-553.

We appreciate the time and effort that you and the reviewers have invested in providing valuable feedback. Please, find below our responses to the Referee's comments.

Yours Sincerely,

Flávio Garcia

REVIEWERS' COMMENTS

Reviewer #1 (Remarks to the Author): I appreciate the significant efforts made by the authors to address the comments provided by reviewers. The manuscript has improved significantly in clarity and experimental details. Adding supporting measurements and more detailed explanations provide a stronger foundation for the work. Considering the improvements made in the manuscript, I think that the manuscript is now suitable for publication.

Response: We thank the Reviewer for this positive feedback.

Reviewer #2 (Remarks to the Author): The authors revised the paper based on the comments by the referees. The revised manuscript is much understandable and persuasive. I recommend the publication.

Response: We acknowledge the Reviewer's positive feedback.

Reviewer #3 (Remarks to the Author): The authors have addressed most concerns raised by the referees. The underlying mechanism for the competition between the metallic and semiconducting states remains ambiguous, the phenomena based on YIG/MoS₂ presented by the authors appear to be robust and repeatable, which probably can apply to other TMD materials. Considering that the substantial impact that this work may bring about to the spintronics field, I can therefore recommend its publication in Nature Communications. Before that, please ensure to correct the following minor typos and add explanations.

1. Following my previous comments on the notation of "A_{1g}", the authors should also correct E' to E_{2g}¹ mentioned in the same caption. Other than the notation issue, the measured two peaks red-shifted by ~6 cm⁻¹ toward lower wavenumbers compared with the literatures. The referee's inference is that the authors didn't do calibration for the MoS₂'s peaks with reference peaks, for instance, peaks from the used substrate. Please explain.

2. In line 239 in Supporting Information, Table S3 seems a typo and should be Table 1.

Response: We appreciate the Reviewer for this positive feedback. The notations were fixed. We agree with the reviewer that the phonon modes (A_{1g} and E_{2g}¹) show a red shift compared to literature mainly for exfoliated MoS₂ samples. We have calibrated our Raman with the

substrate's Silicon peak at 520 cm^{-1} . This red shift is probably due to strain between sample and substrate surface. Tensile strain occurs because of substrate interaction or synthesis conditions that can cause a red shift in Raman peaks. This is caused by stretching the lattice, which reduces the phonon frequencies. CVD grown samples can have residual stress due to growth conditions (temperature/pressure), see for instance reference [1]. The above information was added in the manuscripts supplementary section.

[1] ACS Appl. Nano Mater. 2021, 4, 4172–4180. <https://doi.org/10.1021/acsnm.1c00491>

Reviewer #4 (Remarks to the Author): I co-reviewed this manuscript with one of the reviewers who provided the listed reports. This is part of the Nature Communications initiative to facilitate training in peer review and to provide appropriate recognition for Early Career Researchers who co-review manuscripts.

Response: We sincerely thank the Reviewer for their contribution.

The manuscript by Rodrigo et al. investigates the disentanglement of edge and 2D states and light-driven spin injection in MoS₂. They observed two contributions to spin pumping: (a) from metallic edge states and (b) from 2D semiconductor states. By varying the thickness of MoS₂, they determined that spin current injection by the 2D semiconductor states is purely an interfacial effect, following the inverse Rashba-Edelstein effect. This finding, however, is not novel and has been reported in several previous works. Additionally, they demonstrate that light-driven spin current injection can manipulate spin pumping depending on the size of the MoS₂ flakes, which affects the edge states. While this manuscript is interesting in its current form, it does not provide a substantial advance required for Nature Communications. The authors may consider the following comments before submitting elsewhere:

1. The introduction part could be improved by providing a more detailed discussion on metallic edge states, as this forms the main result of the work. The other parts of the introduction could be made more concise.
2. Have the authors tried preparing samples with metallic ferromagnets instead of YIG and measuring spin-to-charge conversion? It would be interesting to see if spin-to-charge conversion follows the V-shaped dependence similar to Fig. 1e.
3. The enhancement in damping due to spin pumping (α_{SP}) shows a finite value for all points in Fig. 1a (edge length dependence) and 1b (area dependence). How can α_{SP} be zero as shown in Fig. 1e? Please cite previous works, if available, showing this effect.
4. The origin of the compensation point requires more discussion. The authors may need to measure more samples near the compensation point to demonstrate that α_{SP} can indeed be zero.
5. Including simulations or theoretical modelling would help in understanding the results.